# The CSTools (v4.0) Toolbox: from Climate Forecasts to Climate Forecast Information

Núria Pérez-Zanón [1], Louis-Philippe Caron [1,2], Silvia Terzago [3], Bert Van Schaeybroeck [4], Llorenç Lledó [1], Nicolau Manubens [1], Emmanuel Roulin [4], M. Carmen Alvarez-Castro [5], Lauriane Batté [6], Pierre-Antoine Bretonnière [1], Susana Corti [3], Carlos Delgado-Torres [1], Marta Domínguez [7], Federico Fabiano [3], Ignazio Giuntoli [3,8], Jost von Hardenberg [3,9], Eroteida Sánchez-García [7], Verónica Torralba [5], Deborah Verfaillie [10]

[1]Barcelona Supercomputing Center (BSC), Barcelona, Spain
[2]Ouranos, 550 Sherbrooke St W, Montreal, Quebec H3A9, Canada
[3]National Research Council of Italy, Institute of Atmospheric Sciences and Climate (CNR-ISAC), Turin, Italy
[4]Royal Meteorological Institute of Belgium, Brussels, Belgium
[5]Fondazione Centro Euro-Mediterraneo sui Cambiamenti Climatici (CMCC), Bologna, Italy
[6]CNRM, Université de Toulouse, Météo-France, CNRS, Toulouse, France
[7]Delegación territorial (DT) Cantabria, Agencia Estatal de Meteorología (AEMET), Santander, Spain
[8]School of Geography, Earth and Environment Sciences, University of Birmingham, Birmingham, UK
[9]Dept. of Environment, Land and Infrastructure Engineering, Politecnico di Torino, Turin, Italy
[10]Earth and Life Institute, Université catholique de Louvain, Louvain-la-Neuve, Belgium

*Correspondence to*: Núria Pérez-Zanón (nuria.perez@bsc.es)

**Abstract.** Despite the wealth of existing climate forecast data, only a small part is effectively exploited for sectoral applications. A major cause of this is the lack of integrated tools that allow the translation of data into useful and skilful climate information. This barrier is addressed through the development of an R package. CSTools is an easy-to-use toolbox designed and built to assess and improve the quality of climate forecasts for seasonal to multi–annual scales. The package contains process-based state-of-the-art methods for forecast calibration, bias correction, statistical and stochastic downscaling, optimal forecast combination and multivariate verification, as well as basic and advanced tools to obtain tailored products. Due to the modular design of the toolbox in individual functions, the users can develop their own post-processing chain of functions as shown in the use cases presented in this manuscript: the analysis of an extreme wind speed event, the generation of seasonal forecasts of snow depth based on the SNOWPACK model and the post-processing of temperature and precipitation data to be used as input in impact models.

## 1 Introduction

### 1.1 The need for climate information

Large multi-model seasonal forecasting systems have been developed in recent years, both from current international research projects and operational programmes. These include, for instance, the C3S multimodel seasonal forecast system (successor to EUROSIP; Vitart et al., 2007; Mishra et al., 2019; Hemri et al., 2020), APEC (Wang et al., 2009; Min et al., 2014), the North-American Multi-Model Ensembles (Kirtman et al., 2014), and the Decadal Climate Prediction Project (CMIP6-DCPP, Boer et al, 2016). In parallel, there has been an increasing demand for reliable climate information and tailored climate services, in particular at the seasonal timescale, as this period coincides with the planning horizon in several sectors of activities (Troccoli et al., 2008). However, large availability of climate data does not automatically imply have access to useful climate information. Indeed, post-processing methods with different levels of sophistication are required to convert climate data into tailored climate information for each application, allowing users and decision-makers to develop and implement strategies of adaptation to climate variability and to guide well-informed decision making. The generation of tailored climate information can be, for instance, the extraction of global data in a particular region of interest, the correction of the systematic errors that prevent the integration of the climate predictions in impact models or the refinement of the coarse resolution of the climate

datasets in order to be representative of the local climate variability. In fact, there is a strong need and interest of reliable seasonal to decadal forecasts in a wide range of socioeconomic sectors such as energy, agriculture, tourism, health, insurance or logistics to name only a few (White et al., 2017). But the specific information needs for assisting decision-making vary strongly, even within the same sector. For instance, a wind farm owner might be interested in estimating the risk of low cash income due to low winds during a given season and plan a reduction in production accordingly. This requires local information of near-surface wind speed, combined with the specific performance specifications of the turbines (i.e. relevant wind thresholds vary across wind farms). On the other hand, a grid operator might require country-aggregate information of temperature extremes as a proxy for anticipating electricity demand and ensuring the balance of supply and demand in the electricity grid. Similarly, for the agriculture sector, the required climate information may depend on the specific culture (e.g.: olive, wine, or wheat) and even on the specific crop variety, since each of these crops may have different phenological evolution, which implies a climate sensitivity to different climate variables and different time periods. This diversity of user needs makes the generation of tailored products costly in time and resources, something that is sometimes known as the last-mile problem of climate services (Celliers et al., 2021). Sharing software tools that provide state-of-the-art methods solving common problems in the creation of specific climate services can alleviate this problem and facilitate the tailoring process with actors in the climate community and beyond (National Academies of Sciences, Engineering and Medicine, 2016).

In this context, a Climate Services Toolbox (CSTools) has been developed to address these needs. The toolbox was designed to include functions for each of the main seasonal forecast post-processing steps, but the methods are also suitable for sub-seasonal and decadal predictions. These forecasts are typically generated by running a forecast system several times using perturbations on the initial conditions and model physics (ECMWF, 2017). Each simulation is then considered a member of the ensemble. These similarities in setups among forecasts of different time horizons generally lead to common requirements in their post-processing steps (Palmer et al., 2008). Such ensembles are generated to account for initial condition and model uncertainty, to make probabilistic statements about the most likely atmospheric state (ECMWF, 2017) and to inform sensitivity studies. However, additional post-processing steps are required to translate the simulations into climate information. First, this post-processing typically requires hindcasts (past forecasts), ideally generated using the same modelling and data assimilation system as is used to generate the real-time forecasts, to correct modelling system inadequacies, and to generate other forecast products (Hamill, 2011).

CSTools targets primarily applied climate scientists or climate services developers that require the use of high-quality climate data (e.g. high-resolution data obtained by applying downscaling methods). These users can handle the tool by themselves, understanding each of the methodologies given the provided documentation and with the support of scientific research publications. The tool is fully transparent since it is open-source, allowing the user to control, understand and even adapt every step of the analysis in depth. While simple examples are given in the package documentation, this manuscript aims to showcase the usefulness of CSTools in the context of advanced state-of-the-art use cases.

**1.2 From climate data to climate information**

There are different forecast post-processing steps necessary to translate climate data into climate information. These steps will vary depending on the applications, but usually fall within the following categories (as illustrated in Fig. 1):

- *Data collection, curation and homogenization*: This includes collection of data from heterogeneous remote data sources, storage and indexing into local or organisation-accessible file systems or servers, and homogenization for all data files to comply with common internal conventions. The complexity of this step can be high particularly if the data sources do not follow community standards. While this step is out of the scope of this manuscript and the

CSTools toolbox, we refer interested readers to the cds-data-downloader (https://earth.bsc.es/gitlab/es/cds-seasonal-downloader) for this purpose (see Appendix A).

- *Data retrieval and formatting:* This task refers to the loading of climate data from files stored in local or organisation-accessible file systems or servers onto the main memory of the processing workstations, as well as the required arrangement and transformations of the different datasets to be intercompared and analysed. This can be a labour-intensive step when trying to combine multiple or diverse datasets such as observations and forecasts from multiple systems or sources. Slight differences in the internal conventions for storing the respective datasets need to be handled, and methods for spatial and temporal data manipulation, such as spatial interpolation methods, are often necessary.

- *Correction methods for forecast calibration*: Calibration is necessary to correct systematic errors, uncover any predictive signal and adjust forecasts to the observational statistical properties in order to be integrated into impact models. These biases originate from the approximate representation of unresolved climate processes in the forecast systems (Marcos, 2016; Van Schaeybroeck and Vannitsem, 2019; Manzanas et al., 2019).

- *Classification methods for multi-model forecast combination or scenario selection*: Combining multiple forecasting systems allows to substantially enlarge the diversity of potential weather situations (Hemri et al., 2020), errors are partially compensated and there is an increase in consistency and reliability (Hagedorn et al., 2005). Scenario selection, on the other hand, may often be useful for communication and information synthesis for specific applications (Ferranti and Corti, 2011).

- *Downscaling:* Climate forecast systems, due to computational limitations, typically provide global seasonal-to-decadal forecasts at a horizontal resolution of ~100 km. Users, however, require information at a finer scale. As such, statistical and stochastic downscaling techniques are commonly used to perform realistic transformations from large to small scales (Maraun and Widmann, 2018, Ramon et al. 2021).

- *Skill assessment:* Estimating the quality of the predictions is essential to understand the limitations of the simulations, to improve the current forecast systems and to provide useful forecast products tailored to several sectors (Merryfield et al., 2020). Skill estimates should be provided together with the forecast products to allow a correct interpretation of the forecasts or the added value of a system with respect to a benchmark.

- *Visualization*: From the climate services perspective, visualization tools are essential to illustrate different aspects of deterministic or probabilistic climate information.

The primary aim of CSTools is therefore to make post-processing methods (i.e. correction methods for forecast calibration, classification methods for multi-model forecast combination or scenario selection, downscaling methods, and visualization tools) available in one coherent framework in order to facilitate analysis or the post-processing of data such that might be required by an impact model. Because additional steps are required, CSTools also includes functions for data retrieval and formatting as well as skill assessment in order to facilitate the use of the toolbox.

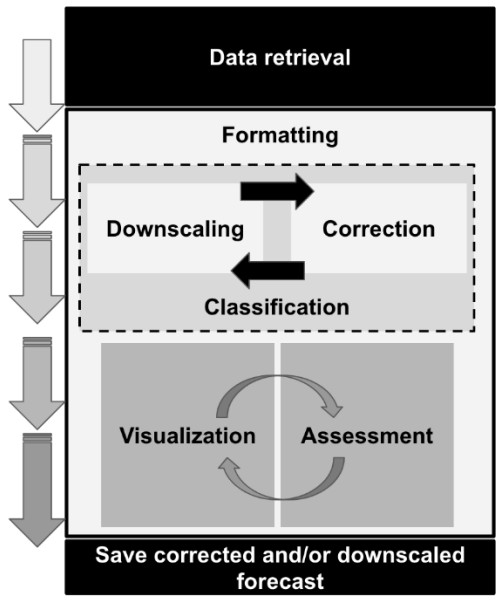

**Figure 1: Scheme of the flexible CSTools workflow (from top to bottom). Each box represents a category of functions that are part of CSTools.**

Several software packages are already available to analyse different types of climate data. For instance, the Earth System Model Validation Tool (ESMValTool; Eyring et al., 2016b; Eyring et al., 2020; Righi et al. 2020) was designed to facilitate the analysis of climate projections produced in the context of the Coupled Model Intercomparison Project (CMIP; Eyring et al., 2016a). The R packages s2dverification (Manubens et al., 2018), SpecsVerification (Siegert, 2017) and easyVerification (MeteoSwiss, 2017) or the python package climpred (Brady and Spring, 2021) focus on skill assessment of ensemble forecasts. Climate4R (Iturbide et al., 2019) is an R based framework for climate data post-processing including different methods. The main purpose of these different packages is the facilitation of research. CSTools, on the other hand, targets scientists interested in providing a climate product to some final users. This is done by allowing the creation of complete post-processing chain from data retrieval to the obtention of high-quality datasets to feed impact models or tailored forecast visualization of forecast products. CSTools could nonetheless be useful to research scientists, as it has been designed to be compatible with some of the aforementioned R packages.

In this manuscript, an overview of the methods and documentation gathered in CSTools is presented in Sect. 2, while the creation of a tailored dataset is shown in Sect. 3. Three case studies based on the analysis of an extreme wind speed event, the snow model SNOWPACK (Lehning et al., 2002a,b https://models.slf.ch/p/snowpack/) and temperature and precipitation data preparation for models requiring evapotranspiration, or similar variables, show the usefulness of the toolbox. Section 4 concludes this paper and discusses some future developments for the package. For a detailed description of CSTools functions and parameters, the reference manual is attached to the package and available at https://CRAN.R-project.org/package=CSTools in the standardized format of an R package documentation.

## 2 CSTools: overview

CSTools was created as part of a collaborative effort between six European institutions. Given the total number of contributors and collaborators (31 in version 4.0), compiling all methods into a software package using the R statistical programming language (R Core Team, 2017) was considered the most suitable and versatile option. Creating an R package allows the inclusion of multiple tools ranging from complex statistical and climatological methods to visualization tools in the same framework. Moreover, CSTools is open-source, thus allowing users and developers alike to benefit from lower costs and software flexibility, quality and reliability (Information Resources Management Association, 2013). At the same time,

CSTools can be integrated into other software in order to take advantage of its functionalities, as does, for instance, the S2S4E Decision Support Tool (https://s2s4e-dst.bsc.es).

CSTools was developed following common guidelines (see supplementary material) agreed upon by all contributors, including conventions for adding new functionalities, and taking into account software development best practices such as the use of a Version Control System (i.e. git; Chacon and Straub, 2014), and testing with continuous integration. The use of these development guidelines has resulted in a clean and homogeneous application programming interface (API).

Most functionalities exposed to the users can be invoked and applied to complex user datasets with a single function call. For example, in order to apply a given functionality named "Func", the user would write:

CST_Func(dataset, ...)

The "CST_*" family of functions ingest and return objects of type "s2dv_cube" (see details in Sect. 2.1), thus allowing
compatibility between each function and long post-processing chains to be created.

## 2.1 Software design aspects of CSTools

The CSTools development guidelines have been designed to maximise compatibility with other libraries such as s2dverification, s2dv, SpecsVerification, easyVerification and startR, all of them designed to operate fundamentally with the same array class. Furthermore, CSTools is also compatible with CSIndicators (Pérez-Zanón et al., 2021) as the latter accepts
"s2dv_cube" objects as inputs. In short, the class "s2dv_cube" is a list of named elements to keep data and metadata in a single object. One of its elements, "data", is a multi-dimensional array with named dimensions containing the values of a variable. The rest of the elements are metadata, such as, spatial coordinates. Because of this design, the CSTools user is able to perform basic array inquiry on the "data" element of the "s2dv_cube" objects at any point in the workflow in order to check the dimensions of the data or to find the number of members, start dates or forecast lead times analysed. Internally, each of these
high-level "CST_*" functions perform two nested calls to two other different but closely related functions in the package. For example, a given functionality named "Func" would involve the following function calls:

```
CST_Func(dataset, ...) {
        ...
Func(dataset$data, ...) {
        ...
                .Func(data_array[i,], ...)
        }
    }
```

At the most fundamental level of this nested call structure, there is a call to a basic function (e.g. ".Func") that is designed to work with the least complex data structure possible (be it a single vector, a couple of vectors, an array and a vector, …). At the second level is a call to a wrapper function (e.g. "Func") around the basic function, which leverages the multiApply package (BSC-CNS et al., 2019) to extend the computation of ".Func" to inputs with any number of dimensions. The top-level "CST_*"
function is an additional wrapper function which adapts the second-level array-based function to work with "s2dv_cube" objects.

This nested structure has several benefits: thanks to the array-compatible low-level functions, the R community can easily employ the CSTools methods in custom CSTools-agnostic workflows if necessary; multi-core parallelism is straightforward

to exploit via middle-level functions and high-level "CST_*" functions; finally, in cases where the data to process is larger than the RAM memory in the workstation or the computation is very expensive, the low-level functions can be used together with the startR package (BSC-CNS and Manubens, 2020) to leverage HPC platforms and distribute workload in small chunks. An example on how to use a CSTools function in a startR workflow can be found in its GitLab repository (https://earth.bsc.es/gitlab/es/startR). Last and foremost, the CSTools user code, using top-level "CST_*" functions, remains modular, concise, readable and easy to maintain. This modular aspect of the "CST_*" functions makes it straightforward for users to create their own post-processing workflows, as shown in Sect. 3. Metadata is propagated and expanded all along the workflows. The development guidelines define conventions to ensure "s2dv_cube" objects are used in a coherent way throughout the package.

## 2.2 Methods in CSTools

Given that the methods included in CSTools are made available as independent functions operating on a common data structure, the users can concatenate them to define their own post-processing workflow. This design provides flexibility allowing the users to assess the impact of the various post-processing steps by modifying the chain of functions. The users can also select a single function and apply it outside of the CSTools workflow. The functions included in the package cover fundamental loading and transformation requirements, downscaling tools, methods for correcting and evaluating forecast and advanced visualization tools (see Table 1). All functions are documented in a standard reference manual on the CRAN website (https://CRAN.R-project.org/package=CSTools). The documentation also includes vignettes, which are self-contained pieces of documentation combining code, text and images, describing some of the methodologies included in CSTools, as well as information on how to use the package to conduct specific analysis.

Table 1. Summary of the functions and methods by category included in CSTools. Prefix "CST_" refers to functions working on a specific object class called "s2dv_cube" while those without the prefix, accept multi-dimensional arrays with named dimensions as input. Asterisks indicates functions that are used in vignettes (see Appendix B for a detailed table).

| Retrieval and formatting | CST_Load*, CST_Anomaly*, CST_SaveExp, CST_MergeDims, CST_SplitDims, as.s2dv_cube, s2dv_cube |
|---|---|
| Classification | CST_MultiEOF, CST_WeatherRegimes*, CST_RegimesAssign*, CST_CategoricalEnsCombination, CST_EnsClustering* |
| Downscaling | CST_Analogs*, CST_RainFarm*, CST_RFTemp, CST_AdamontAnalog, CST_AnalogsPredictors |
| Correction | CST_BEI_Weighting*, CST_BiasCorrection, CST_Calibration, CST_QuantileMapping, CST_DynBiasCorrection |
| Assessment | CST_MultiMetric*, CST_MultivarRMSE* |
| Visualization | PlotCombinedMap*, PlotForecastPDF*, PlotMostLikelyQuantileMap*, PlotPDFsOLE, PlotTriangles4Categories* |

### 2.2.1 Retrieval and transformation function

CSTools includes a function to retrieve data from netCDF files called *CST_Load*. This function is a wrapper of the s2dverification *Load* function which allows to load monthly or daily forecast data together with date-corresponding observations (Manubens et al., 2018). The function allows to easily combine subsets of data stored in multiple files in POSIX

file systems or OPeNDAP servers, and is designed to support custom conventions for distribution of data across files, file

naming, and NetCDF structure. Optionally, CSTools can automatically interpolate all the data onto a common grid if necessary, thus greatly removing complexity for the user. Three samples of "s2dv_cube" objects created from using *CST_Load* are provided along with the package: *area_average*, with forecast and observational climate data averaged over a region; *lonlat_data* and *lonlat_prec* containing forecast and observational climate data for temperature and precipitation.

For users who retrieve data by other means (e.g. using the library ncdf4; Pierce, 2019), the CSTools package contains two functions to convert data to a "s2dv_cube" object. If the data and metadata have been loaded in separate objects, they can be merged into a "s2dv_cube" object with the function *s2dv_cube*. On the other hand, if the data and metadata have been loaded into a single object, it can be transformed into class "s2dv_cube" class with the *as.s2dv_cube* function.

One of the capabilities of CSTools is to create a new dataset after, for example, the data has been downscaled and/or calibrated. In that case, the user may need to save the new dataset into files to be shared among other users or its community. Therefore, the package comes with a saving function called *CST_SaveExp* which creates netCDF files in a directory set by the user and which can be loaded again with the *CST_Load* function. Moreover, the climatological essential steps of computing anomalies can be done with *CST_Anomaly* which is a wrapper function of s2dverification methods that also allows computing smoothed

climatologies.

The functions *CST_MergeDims* and *CST_SplitDims* provide additional flexibility to manipulate "s2dv_cube" objects. For instance, it is commonly required to split the time dimension of annual data into two dimensions, one identifying the season and the other the month of that season. On the contrary, some advanced classification methods may need to merge the

latitudinal and longitudinal coordinates in a single dimension.

### 2.2.2 Classification methods

Classification methods are widely used in climatology to summarize the climatological conditions captured by observations or simulations. Sokal (1966) was already using sophisticated univariate and multivariate climatic classification systems to be generated from enormous data bases (Balling, 1984). However, the functions included in CSTools for this purpose are modern

methods adapted to observations, reanalyses and climate model outputs with multiple ensemble members.

The *CST_MultiEOF* function allows conducting Empirical Orthogonal Functions (EOF) analysis simultaneously over multiple variables for either each ensemble member or all the ensemble members concatenated altogether (i.e., it can be applied to each one of the ensemble members separately or to the whole ensemble). Based on singular value decomposition, the EOF analysis

is applied over the region of interest (for example the Mediterranean region) in order to define, for each of the N variables chosen, a reduced phase space based on the leading modes of variability. A simultaneous analysis of these fields is then carried out with a (multivariate) EOF analysis in the subspace spanned by the leading EOFs of each field. This produces a N-variable EOF picture of the variability in the region. The associated principal components can represent multi-variable indices that can be used to verify the forecast.


*CST_WeatherRegimes* and *CST_RegimesAssign* are complementary functions to derive weather regimes (Cortesi et al., 2019; Torralba, 2021). The first function computes a set of weather regimes using a cluster analysis. The dimensionality of this object can also be reduced by using PCs obtained from the application of the EOF analysis to filter the dataset, while the cluster analysis can be performed with the traditional k-means or hierarchical clustering. On the other hand, *CST_RegimesAssign*

matches anomalies to a set of reference maps obtained using *CST_WeatherRegimes*. The anomalies are assigned to the most

similar reference map using either the minimum Euclidean distance or the highest spatial correlation, which can be particularly useful to classify the predictions according to the clusters identified in the observational reference.

*CST_CategoricalEnsCombination* converts a multi-model ensemble forecast into a categorical forecast by giving the probability for each category. Different methods are available to combine the different ensemble forecasting models into probabilistic categorical forecasts. The user can set up the total number of categories that will be used to define the observed climatological quantiles. The available methods are: "pool" for ensemble pooling where all ensemble members of all forecast systems are weighted equally; "comb" for a model combination where each model system is weighted equally; and "mmw" for model weighting. The model weighting method is described in Rajagopalan et al. (2002), Robertson et al. (2004) and Van Schaeybroeck and Vannitsem (2019). More specifically, this method uses different weights for the occurrence probability predicted by the available models and by a climatological model and optimizes the weights by minimizing the ignorance score which is a measure of the information conveyed by a forecast (Tödter and Ahrens, 2012).

*CST_EnsClustering* is a cluster analysis tool, based on the k-means algorithm, for ensemble predictions. The aim is to group ensemble members according to similar characteristics and to select the most representative member for each cluster. The user chooses which feature of the data is used to group the ensemble members by clustering (e.g. temporal mean). The anomaly is computed with respect to the ensemble members and the EOF analysis is applied to these anomaly maps. After reducing dimensionality via EOF analysis, k-means analysis is applied using the desired subset of PCs. The user can choose how many Principal Components (PCs) to retain or the percentage of explained variance to keep for the EOF analysis.

### 2.2.3 Downscaling methods

Downscaling is designed to increase the resolution of a dataset. In a climate service chain, downscaling is a fundamental step to transform the climate simulations from their coarse resolution to the finer resolution required by many final users studying regional environmental changes (Maraun et al. 2010; Rössler et al., 2019). CSTools contains five different downscaling methodologies based on analog techniques, stochastic simulations or regression.

The *CST_Analogs* function can be used to downscale any gridded dataset using analogs. The function, based on the method of Yiou et al. (2013), searches for days with similar large-scale conditions to provide high-resolution fields over a specific region. Regions and variables can be defined by the user and three different criteria to select the analogs are available: (1) minimum Euclidean distance in the large-scale pattern, (2) minimum Euclidean distance in a large-scale pattern and in a local-scale pattern, and (3) minimum Euclidean distance in a large-scale pattern and a local scale pattern as well as maximum correlation in a local variable to be downscaled. Typically, criteria (1) is used to find the analog based on a large-scale variable (e.g. sea level pressure/geopotential in the North Atlantic or sea surface temperature over the tropics). Criteria (2) helps to confirm that both large-scale patterns and the large-scale variable in a local scale (e.g. sea level pressure in the Iberian Peninsula) are consistent. Criteria (3) also measures the similarities between the large-scale variable and a different variable (e.g. surface temperature in the Iberian Peninsula) in the local scale.

*CST_RainFARM* implements a stochastic downscaling technique and represents a so-called full-field weather generator. More specifically, this function generates synthetic fine-scale precipitation fields whose statistical properties are consistent with the small-scale statistics of observed precipitation, while preserving the properties of the large-scale precipitation field. The Rainfall Filtered Autoregressive Model (RainFARM; Rebora et al. 2006a,b) is based on the nonlinear transformation of a linearly correlated stochastic field generated by small-scale extrapolation of the Fourier spectrum of a large-scale precipitation field. Developed originally for downscaling data at weather timescales, the method has been adapted for downscaling at climate

timescales by D'Onofrio et al. (2014) and recently improved for regions with complex orography, for which the fine-scale fields produced by RainFARM are corrected using weights derived from a fine scale precipitation climatology (Terzago et al., 2018). This methodology relies on two distinct functions to compute weights from high-resolution climatologies (*CST_RFWeights*) and the spatial-spectral slope used to extrapolate the Fourier spectrum to the unresolved scales (*CST_RFSlope*).

*CST_RFTemp* implements a simple lapse rate correction to a near-surface temperature field to account for changes in orography between a low- and high-resolution gridded dataset.

ADAMONT (ADAptation of RCM outputs to MOuNTain regions; Verfaillie et al., 2017) is a downscaling method designed to adjust forecasts of daily variables. The method is based on the quantile mapping approach and originally relied on a regional reanalysis of hourly meteorological conditions. Two functions to implement ADAMONT have been included in CSTools. *CST_AdamontQQcor* computes a quantile mapping based on weather types for forecast data while *CST_AdamontAnalog* uses these weather types to find analogous data in the reference dataset.

The *CST_AnalogsPredictors* function downscales precipitation or maximum/minimum temperature low resolution forecast output data through the association with an observational high-resolution dataset (Peral García et al., 2017) and a collection of predictors and reference synoptic situations similar to the estimated day. As a first step, a partner function *AnalogsPredictors_train* must be run to compare the large-scale atmospheric circulation to each of the atmospheric configurations from a reference period. The most similar days, defined by the Euclidean distance of winds, are chosen as their analogs.

### 2.2.4 Correction methods

Correction methods can improve the quality of simulations by reducing the systematic errors that are present in the forecast due to model deficiencies. The periodicity of modes of variability (i.e. space-time patterns that tend to recur in the observed record) can also be exploited to improve the forecast skill.

Sánchez-García et al. (2019) used the North Atlantic Oscillation (NAO) to improve the skill of the seasonal precipitation forecast over the Iberian Peninsula. Given that this methodology could be explored to improve the skill of different climate variables that are led by other climate indices, the method has been generalized and named Best Estimate Index (BEI). The methodology consists of three functions: *BEI_PDFBest* combines the climate indices from the two forecast systems, *BEI_Weights* provides the weights to correct a forecast system and *CST_BEI_Weighting* computes the ensemble mean or the tercile probabilities considering the weights returned by *BEI_Weights*.

Calibration can be considered as a way of obtaining predictions with average statistical properties similar to those of a reference data set. *CST_Calibration* performs the correction on the forecast systems' simulations using five different member-by-member methodologies, where each methodology can adjust one or more statistical properties of the predictions. The selection of the most appropriated method will thus depend on the user's needs. The "bias" method corrects the mean bias only, the "evmos" method applies a variance inflation technique to ensure the correction of the mean and the correspondence of variance between forecasts and observations (Van Schaeybroeck and Vannitsem, 2011). The ensemble calibration methods "mse_min" and "crps_min" correct the bias, the overall forecast variance and the ensemble spread as described in Doblas-Reyes et al. (2005) and Van Schaeybroeck and Vannitsem (2015), respectively. While the "mse_min" method minimizes a constrained mean-squared error using three parameters, the "crps_min" method features four parameters and minimizes the Continuous

Ranked Probability Score (CRPS). The "rpc-based" method adjusts the forecast variance to ensure that the ratio of predictable components (RPC) is equal to one (Eade et al., 2014). The function allows the five calibration methods to be performed in leave-one-out cross-validation mode, which means that the observed value of the year that is being corrected is not considered in the calibration, as it would be the case for real-time forecasts (Doblas-Reyes et al., 2005, Torralba et al., 2017). The use of cross-validation is particularly important in order to avoid overestimating the skill when the hindcasts are calibrated. *CST_BiasCorrection* performs the same analysis as *CST_Calibration* using the "evmos" method but allowing to calibrate either a hindcast or forecast.

*CST_QuantileMapping* performs a quantile mapping adjustment by matching the probability distribution of a forecast with the probability distribution of a set of observations. The function in CSTools calculates the relation between a set of past forecasts (i.e. hindcasts) and observations and applies the correction to the hindcast itself or to a different forecast. This function relies on the R package qmap (Gudmundsson et al., 2012; Gudmundsson, 2016). The user can set several parameters to define the distance between quantiles when adjusting the distribution, or the sample length in cases when the user wants to split the temporal dimension to apply separate adjustments.

*CST_DynBiasCorrection* relies on the dynamical state of the system to correct the systematic errors rather than on its statistical properties. This method uses two dynamical system metrics to correct the bias of each ensemble member: the local (in phase space) dimension and the persistence. In simple terms, they describe the recurrences of a system around a state in phase space. Dimension provides information on how the system can reach a state and how it can evolve from it. Thus, dimension is a proxy for the system's active number of degrees of freedom. A very persistent state is typically highly predictable, while a very unstable state yields low persistence (Faranda et al., 2017; Faranda et al., 2019). The functions *CST_ProxiesAttractor* (to compute local dimension *d* and inverse of persistence *theta*) and *Predictability* (to compute scores of predictability based on the dynamical indicators resulting from *CST_ProxiesAttractor)* are internally used by *CST_DynBiasCorrection* and they are also exposed for users interested in interpreting the method's intermediate results.

**2.2.5 Verification functions**

Verification is not the main objective of this package. For that purpose, we refer users to other R packages such as s2dverification, SpecsVerification and easyVerification. However, in order to facilitate the evaluation of the forecasts, some basic metrics have been included.

*CST_MultiMetric* calculates correlation, root mean square error and the root mean square error skill score for individual models and multi-model mean (if desired; Mishra et al., 2019) as well as the ranked probability skill score (RPSS) based on terciles.

*CST_MultivarRMSE* calculates the RMSE using multiple variables simultaneously. The output is the mean of each variable's RMSE scaled by its observed standard deviation. Variables can also be weighted based on their relative importance (as defined by the user).

**2.2.6 Visualization**

Some of the most requested functionalities in climate services are data visualization tools that allow presenting large quantities of information in an intuitive way. All the visualization functions in CSTools can be customized by modifying colours, titles, sizes, etc. and it is possible to save them to files in different formats (e.g. .ps, .eps, .png, pdf, ...) or display the result in a pop-up window.

*PlotCombinedMap* combines multiple 2-dimensional datasets into a single map based on a decision function. In other words, several "maps" are provided as input, and for each "map" the function creates a colour legend. A decision function is used at each gridpoint to choose the value to be displayed, in the process retaining the information of which "map" it belongs to. For instance, multiple model skills could be compared in a region to visualize which is the best model in each region (Fig. 2; Mishra et al., 2019). Other applications, such as comparing multiple variables, are also possible.

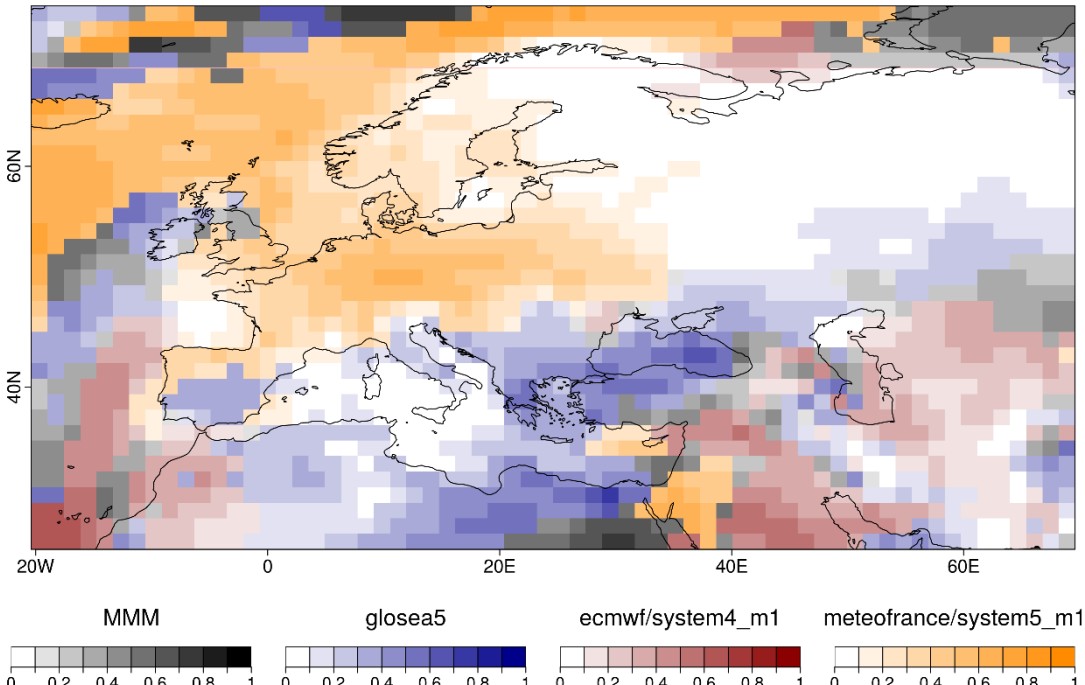

**Figure 2: Example of Plot Combined Map result over Europe comparing different forecasts systems and the multi-model mean by colours and the highest correlation value in each grid point by colour intensity.**

*PlotMostLikelyQuantileMap* allows visualizing different probabilities easily. It receives as main input (via the parameter "probs") a collection of longitude-latitude maps, each containing the probabilities (from 0 to 1) of the different grid cells belonging to a category: terciles, quantiles, or others (Fig. 3; Lledó et al., 2020a; Torralba, 2019). The function plots the probability for the category with the maximum probability for each grid point.

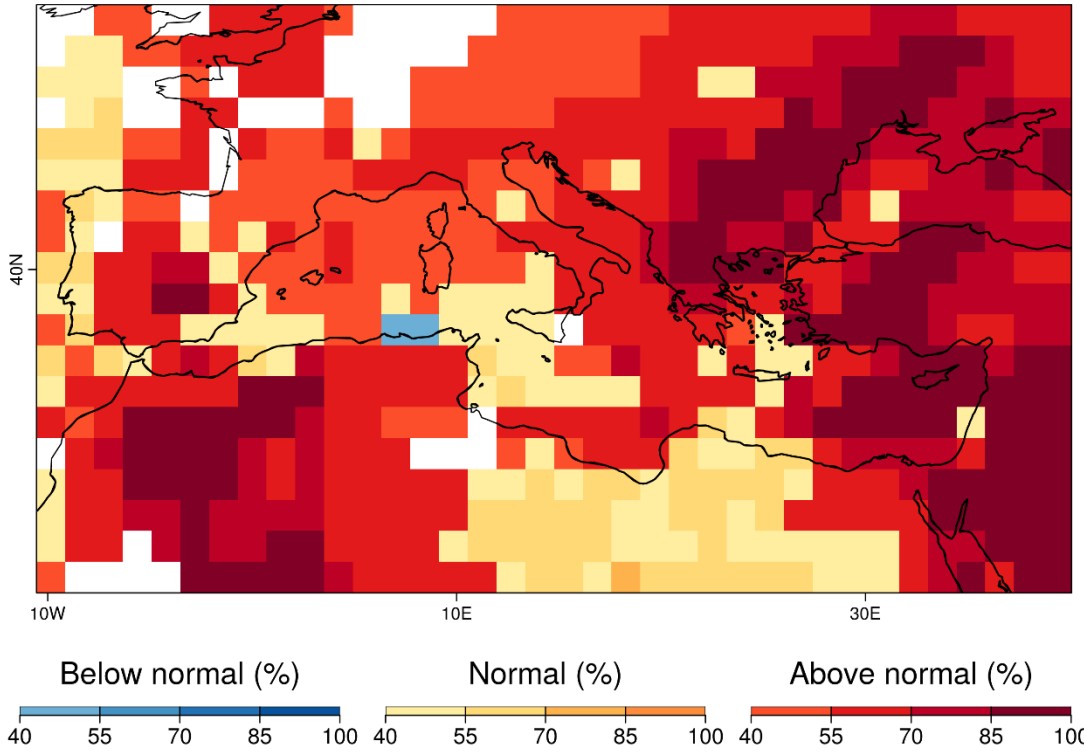

**Figure 3: Example of PlotMostLikelyQuantileMap for which the tercile probabilities of the summer sea surface temperatures of 2020 in the Mediterranean region are compared. The function allows to apply a mask (white grid points) which in this case correspond to areas where the system doesn't have sufficient skill for a given metric over the verification period.**

*PlotForecastPDF* plots the probability distribution function of several ensemble forecasts in separate panels. By default, the function plots the ensemble members, the estimated density distributions and the tercile probabilities (Fig. 4). The density functions are approximated by dressing the ensemble members with a kernel density estimate technique using a gaussian kernel (Bröcker and Smith, 2008). Silverman's rule of thumb is used to select the spread of the kernel, which controls the degree of

410 smoothing. Probabilities for extreme categories, above (below) the 90[th] (10[th]) percentile (from now on, P90 (P10)), and observed values can also be included. This function is useful to compare changes in forecasts with different lead times (Soret et al., 2019). A comparison between forecasts from different models, different modes of variability (Lledó et al., 2020a) or even forecasts at different locations is also possible.

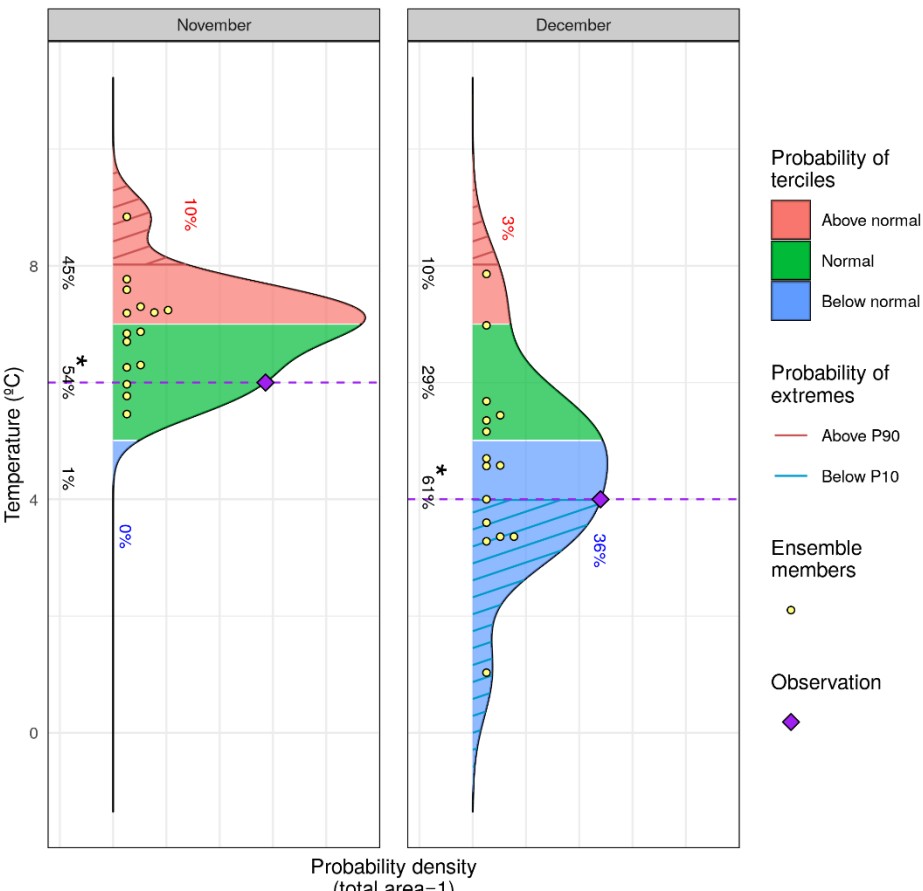

Forecasts initialized on Nov 2000 at sample Mediterranean region

**Figure 4: Example of PlotForecastPDF result using the *lonlat_data* sample provided along with CSTools showing a possible comparison of two different lead times for a given forecast in a specific location. The individual ensembles (yellow dots), the corresponding observation (purple diamond), the probability of each tercile, i.e.: above normal (pink), normal (green) and below normal (blue); are annotated and the one with highest probability is marked with an asterisk and the extreme limits, i.e. above 90 percentile (red) and below 10 percentile (blue) correspond to the hatched areas and are also annotated.**

*PlotPDFsOLE* plots two probability density Gaussian functions and their combination by the optimal linear estimation (OLE; Fig. 5). The mean and the standard deviation of the two probability functions must be provided (Sánchez-García et al., 2019).

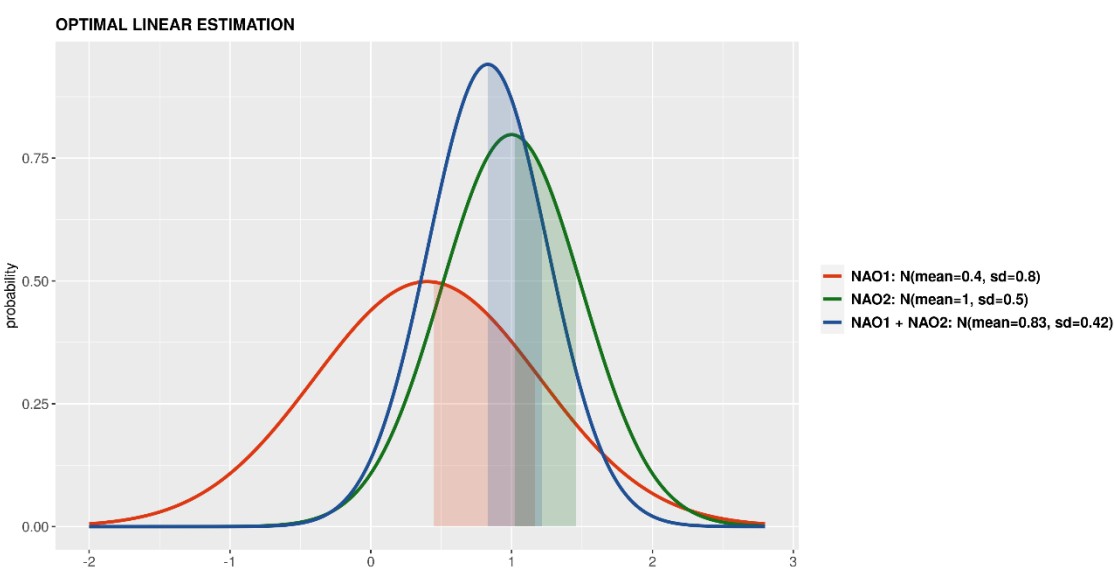

**Figure 5: A synthetic example of PlotPDFsOLE in which two distribution function are combined.**

It can sometimes be useful to present tabular results as colours instead of numbers. For this purpose, *PlotTriangles4Categories* converts a 3-dimensional numerical data array into a coloured grid with triangles (Fig. 6). This function can be used to quickly compare modes of variability, skill metrics, differences between methods or forecast systems as a function of the lead times or seasons (Torralba, 2019; Verfaillie et al., 2021; Lledó et al., 2020b).

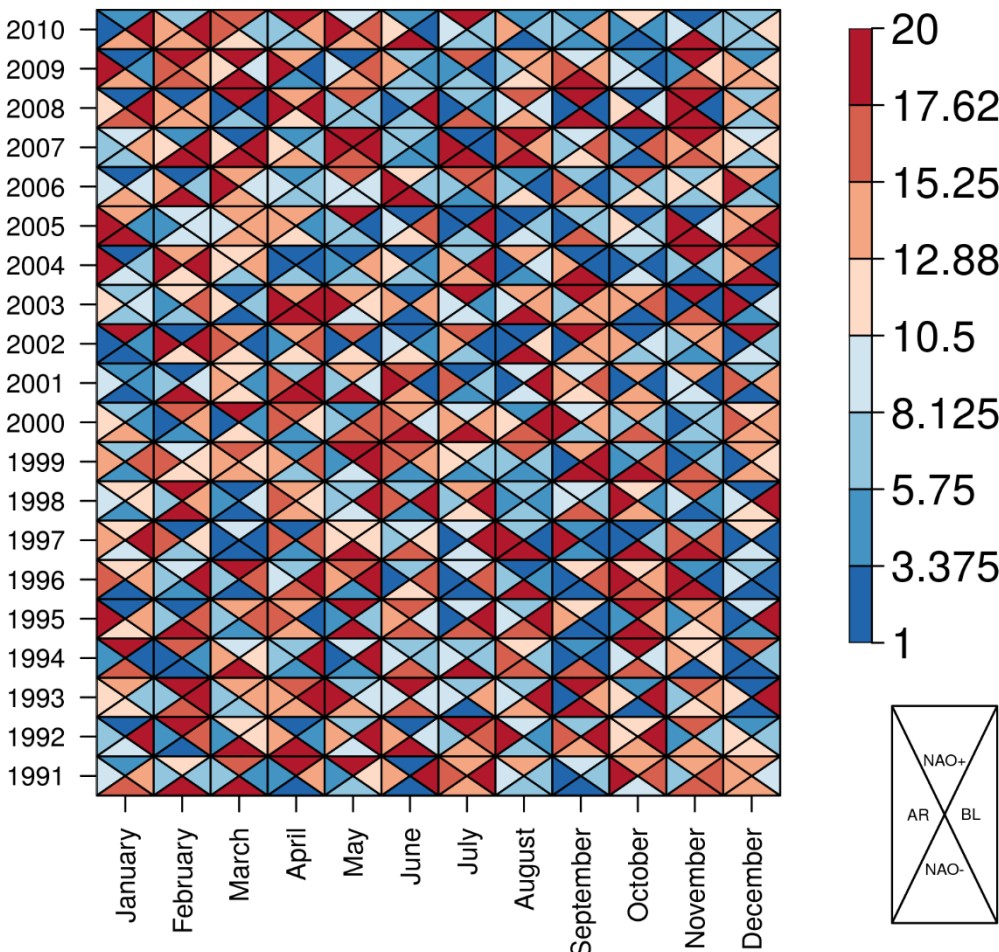

**Figure 6: Example of the PlotTriangles4Categories using 4 different categories.**

Real examples of these visualisation tools as well as other functions of the package are shown through the three example case studies provided in the next section.

**3 Use Cases**

In order to demonstrate how CSTools can be used to provide climate information to potential users, we present three case studies which rely on CSTools for data post-processing. The first case study assesses whether seasonal forecasts could anticipate the very strong near-surface winds over the Iberian Peninsula in March 2018 such as to provide useful information to the energy sector. In the second case, seasonal forecasts of precipitation are post-processed following the requirements to use them to drive model of snowpack depth in high mountain sites. Finally, we provide an example of how seasonal forecasts

of rainfall and near-surface temperature can be post-processed.

**3.1 Use case 1: Assessing the odds of an extreme event**

In March 2018, the Spanish Meteorological Agency activated its protocol of early warning system for 47 regions of Spain due to the high-speed winds forecasted and possible coastal impact. Very high wind speeds were later recorded over large parts of

the Iberian Peninsula due to the passing of four cyclones (AEMET, 2018). This type of event is of interest to the energy sector, given its impacts on wind power generation, energy demand and electricity prices, and such interest is likely to keep rising as we continue transitioning towards, and become more reliant on, renewable energy. For context, the renewable energy production had grown substantially in Spain over the course of 2017-18: renewable energy generation was 51.1 % higher in March of 2018 compared to what it had been during the same month of the previous year. A historical maximum of monthly renewable generation was hit with 13,204 GWh (33.1 % of share), of which wind energy contributed 7,676 GWh, setting also a new record of monthly wind generation (Red Eléctrica España, 2018). These high amounts of renewable generation in March 2018 resulted in an important drop in electricity prices. Because of its strong impact on the market, there is a lot of interest in the energy sector to anticipate this type of events.

The use case presented here shows whether the 2018 event had been anticipated a few months in advance by a seasonal forecast system. For both the hindcasts and the forecasts, we use monthly means of 10 m wind speed from the latest ECMWF long-range forecasting system SEAS5, obtained from the C3S datastore (SEAS5; Johnson et al. 2019) at 1° spatial resolution. For the observational reference, we use monthly mean 100 m wind speeds from the ERA5 reanalysis (Hersbach et al., 2020) at 0.25° (around 30 km) spatial resolution. The seasonal forecasts initialized on December 2017, January 2018 and February 2018 are assessed. For each start date, three different data types are required: the hindcast, i.e. retrospective forecasts initialized in the past for start dates ranging from 1993 to 2016 for December initialization and 2017 for January and February initializations; the observational reference, covering the same period as the hindcast; and the operational forecast, i.e. the latest simulations initialized just before the event (i.e. December 2017, January 2018 and February 2018).

Two functions from CSTools are used to post-process the wind speed seasonal forecasts: *CST_Load* and *CST_BiasCorrection*. The key decisions are the parameters used to retrieve the data from files to achieve a coherent analysis of the March 2018 event (Fig. 7). The analysis is repeated for three different start dates (i.e. December, January and February). In all the data loading calls, the same region must be requested through the parameters *lonmin*, *lonmax*, *latmin* and *latmax* of the function *CST_Load*. The output type requested to be gridded data rather than area average by setting *output* parameter as "lonlat". Note that the code could be adapted to other regions, time periods and variables and a detailed description of the code is provided below for users interested in modifying the necessary parameters.

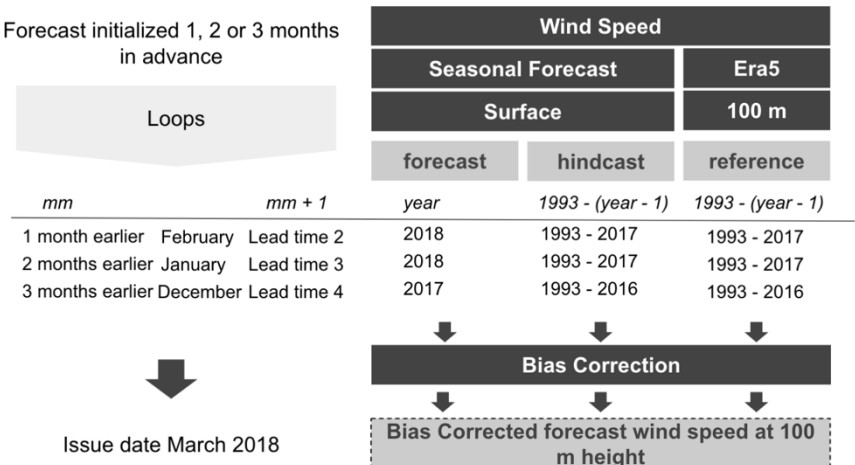

**Figure 7: Scheme of the methodology applied. Grey boxes indicate the data, methods and results. The required parameters to analyse the March 2018 event are specified for the simulations initialized one, two and three month(s) in advance on white background.**

The name of the different variables required must be specified in the *CST_Load* call through the parameter *var*, and the function will read the corresponding variables written in the NetCDF files in the data storage. Therefore, the *var* parameter is set to "sfcWind" when retrieving hindcasts and forecasts, while for the reference dataset it is set to "windagl100". Given the

difference in spatial resolution, a regridding of the reference dataset is also requested via the *grid* parameter. The path patterns

expressing the set of files that comprise the simulations and the reference datasets are also passed to the *CST_Load* function

through parameters *exp* and *obs*, respectively. Notice that the labels $STORE_FREQ$, $VAR_NAME$, $START_DATE$, $YEAR$ and $MONTH$ are used when defining the patterns. These labels will be interpreted and replaced by the function following the information provided in the other parameters of *CST_Load*.

An index *mm* indicating the number of preceding months (*mm*) is introduced to loop over the three start dates in order to simplify the code. When *mm* is 1, the bias adjustment for March with the forecasts initialized one month in advance (i.e. the February start date) is computed. The target year is set in the "year" variable as 2018. The start dates of the simulations to be loaded are created and stored in the "hcst_sdates" and "fcst_sdates" variables, which correspond to a vector of dates for the 1st of February from 1993 to 2017 and the 1st of February 2018, respectively. For the February start date, the lead time two

(i.e. *mm* + 1) corresponds to the forecast for March which is selected through the *leadtimemin* and *leadtimemax* parameters.

Finally, a simple bias correction method (*CST_BiasCorrection*) is used to compute the biases between the hindcast and the reference datasets and then apply a correction to the mean and standard deviation of the forecast dataset. The results of each loop are stored in a list. Winds at 100 m height are of relevance for energy applications and, although this variable is not

available directly from the seasonal prediction system, the bias adjustment procedure will convert 10 m to 100 m winds by assuming a logarithmic wind profile (Drechsel et al., 2012).

```r
library(CSTools)
exp_path <- list(name = "ECMWFS5",
        path =
"/esarchive/exp/ecmwf/system5c3s/$STORE_FREQ$_mean/$VAR_NAME$_f6h/$VAR_NAME$_$START_DATE$.nc")
obs_path <- list(name = "ERA5",
        path =
"/esarchive/recon/ecmwf/era5/$STORE_FREQ$_mean/$VAR_NAME$_f1h/$VAR_NAME$_$YEAR$$MONTH$.nc")
 # Target months March (3)
 # Assess forecast from 1 to 3 months in advance
months_in_advance <- c("02", "01", "12")
for (mm in 1:3) {
 # Generate the start dates of hindcast period
 year <- ifelse(mm == 3, 2017, 2018)
 hcst_sdates <- paste0(1993:(year - 1), months_in_advance[mm], "01")
  # Load hincast data
 wind_hcst <- CST_Load(var = "sfcWind", exp = list(exp_path),
                        sdates = hcst_sdates, nmember = 25,
                        leadtimemin = mm + 1, leadtimemax = mm + 1,
                        storefreq = "monthly", sampleperiod = 1,
                        latmin = 36, latmax = 44, lonmin = -10, lonmax = 4,
                        output = "lonlat")
 # Generate the start dates of forecast period
 fcst_sdates <- paste0(year, months_in_advance[mm], "01")
 # Load forecast data
 wind_fcst <- CST_Load(var = "sfcWind", exp = list(exp_path),
                        sdates = fcst_sdates, nmember = 25,
                        leadtimemin = mm + 1, leadtimemax = mm + 1,
                        storefreq = "monthly", sampleperiod = 1,
                        latmin = 36, latmax = 44, lonmin = -10, lonmax = 4,
                        output = "lonlat")
 # Load reference data
 wind_ref <- CST_Load(var = "windagl100", obs = list(obs_path),
                        sdates = hcst_sdates, nmember = 1,
                        leadtimemin = mm + 1, leadtimemax = mm + 1,
                        storefreq = "monthly", sampleperiod = 1,
                        latmin = 36, latmax = 44, lonmin = -10, lonmax = 4,
                        output = "lonlat",
                        grid = "r360x181")
 # Bias Adjustment
 wind_fsct <- CST_BiasCorrection(exp = wind_hcst,
                        obs = wind_ref,
                        exp_cor = wind_fcst)
```

Once the forecasts are post-processed, additional CSTools functions can be used to visualize the forecast distributions. The *PlotForecastPDF* function, for instance, compares the probability distribution function of the March 2018 100 m wind speed

forecasts issued 1 to 3 month(s) in advance (Fig. 8). Three months in advance, only one member out of 25 exceeds the P90. The simulations initialized one and two month(s) in advance suggest a weak shift towards above-normal conditions (~40 % probability of the above normal tercile) and towards extreme high values (12 % and 17 % exceeding P90). Moreover, the forecast's tercile probabilities do not indicate a shift towards above-normal winds as lead time decreases (the January start date suggests a slightly larger probability of above-normal winds than the February start date). Even though for start dates in

both January and February three members exceed P90, the corresponding probabilities are different due to the ensemble dressing applied (see 2.2.6 for details). In February, the probability of observing extreme wind conditions was almost twice as large as in January. Individual ensemble members typically also suggest much weaker wind speed anomalies than observed, except for one member in the February initialization, indicating that in this case the prediction system anticipated this situation as a potential outcome.

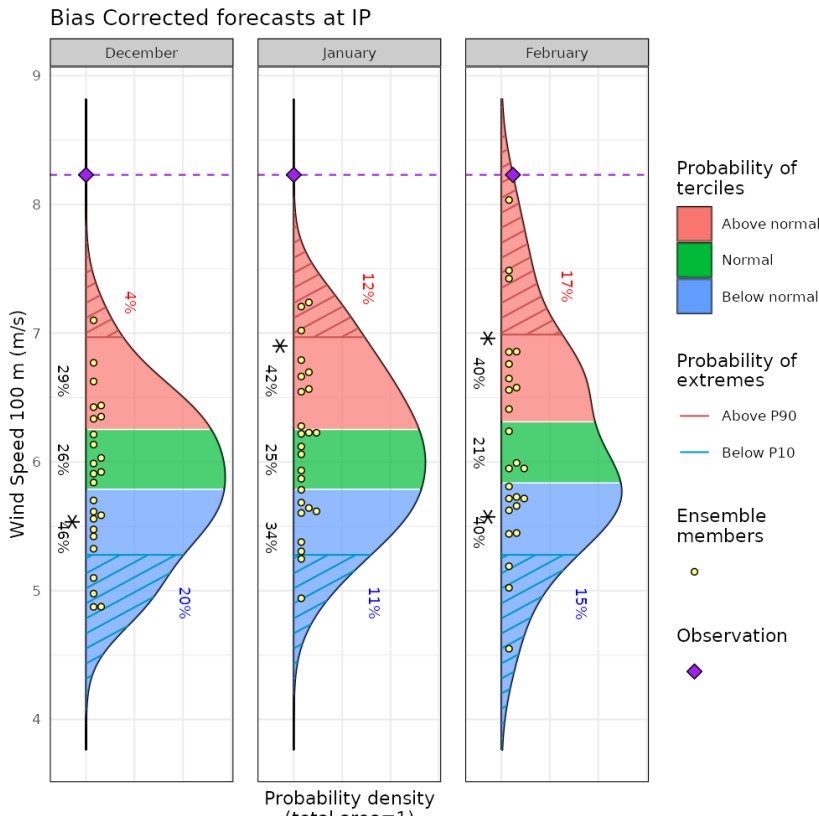


**Figure 8: Seasonal forecasts of wind speed at 100 m height, averaged over 10 ºW-4 ºE and 36-44 ºN for March 2018. Each panel corresponds to forecasts launched 3 to 1 month(s) ahead (from left to right). Methodology: simple bias correction with ERA5 observations, based on previous hindcasts since 1993. An asterisk indicates the tercile with the highest probabilities.**

The spatial distribution of the tercile probabilities can be displayed with the *PlotMostLikelyQuantileMap* function (Fig. 9). An extra layer has been included to mark with crosses the grid points where observations agree on the most likely tercile indicated by the forecast. Three months in advance, most of the region shows that the tercile of highest probability is the below-normal category. One and two month(s) in advance, the colours shift towards the normal and above-normal categories. In the January simulation, the eastern region presents more above-normal probability of high wind speed values than the western region. In

the February simulation, the above-normal probability class is widespread on the whole Iberian Peninsula.

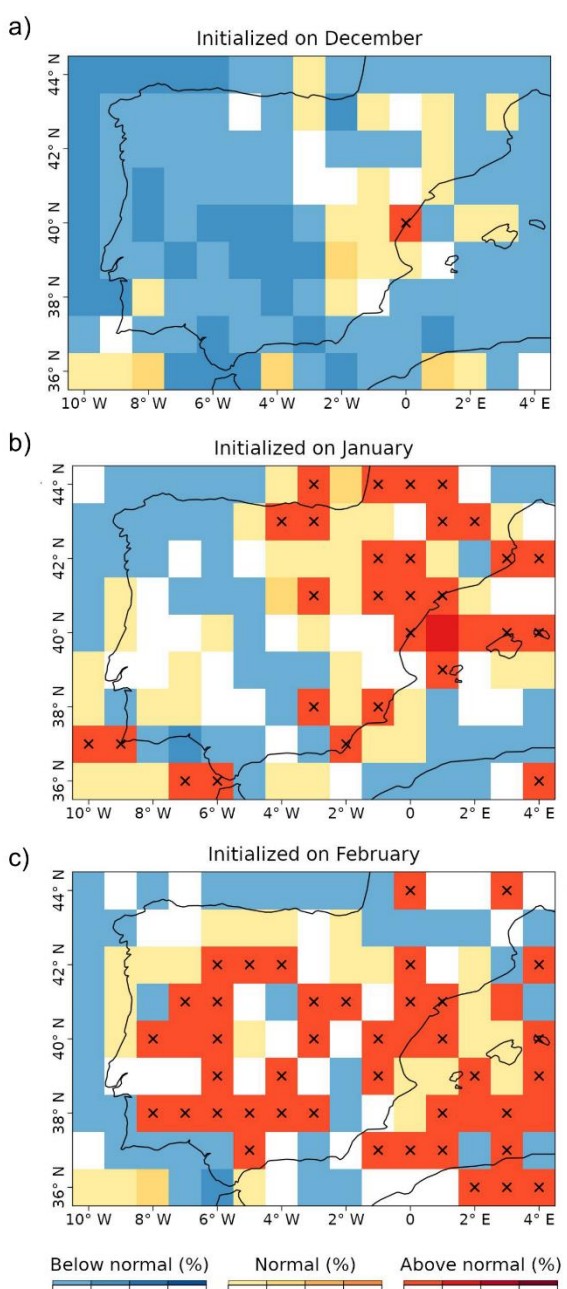

**Figure 9: Probabilities of the most likely tercile for the March 2018 100 m wind speeds, as indicated by the forecasts issued 3 to 1 month(s) ahead (top to bottom). The crosses indicate that the observations fell into the most likely tercile displayed by the forecast. White grid points indicate that no tercile category has more than 40 % of probability.**


Users that can benefit from climate information, such as stakeholders (e.g. energy system planners), are usually not familiar with probabilistic forecasts and the added value that it could potentially bring to their planning. In order to become more autonomous in their decision making, a learning process could be started based on relevant show-case climate events such as the one provided here. Therefore, such use case could be of interest for climate services developers who need to post-process

a seasonal forecast variable and present the results in a concise yet user-friendly manner with a reduced number of images and tables.

The code of this use case could be reused to evaluate the seasonal forecast performance. The evaluation could consider whether other variables were better capturing the situation in a specific event, or if a different bias correction method would improve

the skill of the seasonal forecast. Furthermore, it is possible to easily modify this code to compare the results provided by different models.

### 3.2 Use case 2: Seasonal forecast of snow depth and snow water equivalent in high-elevation sites

The post-processing of seasonal precipitation forecast in the Alps to be used as input for the SNOWPACK model is shown in this use case, as well as the result of the SNOWPACK model snow depth. The relevance of this use case is that alpine snowpack in the mountains represents an essential water reservoir that is fed by snowfall during the cold season and then released in late spring and summer. Mountain meltwater is essential for several economic activities including hydropower generation, agriculture and industry, while meltwater shortage can induce strong economic losses. Therefore, reliable seasonal forecasts of snow resources that, at the beginning of the snow season (November), estimate the snow accumulation towards the end of spring (April-May), are highly pursued. These would allow water management authorities and hydropower companies to implement early water management plans several months ahead of a water-demand peak and mitigate the effects of a possible water shortage. To support this need, a modelling chain driven by seasonal forecasts of essential climate variables from the C3S seasonal forecasting systems was developed, employing the physical 1-dimensional snow model SNOWPACK (Bartelt and Lehning, 2002), to estimate snow depth and snow water equivalent at selected high-elevation sites in the North-Western Italian Alps.

The SNOWPACK model requires a number of input variables, namely 2 m air-temperature, atmospheric pressure, relative humidity, shortwave and longwave incoming radiation, wind speed and ground temperature, at finer spatial and temporal resolutions (hourly, 1 km data) compared to the typical resolutions of the seasonal forecast system outputs (6 hourly or daily, about 100 km data). In order to provide the SNOWPACK model with the required climatological forcing, we apply bias-adjustment and a downscaling techniques depending on the specific variable.

- Seasonal precipitation forecasts need to be downscaled and bias-corrected.
- Seasonal temperature forecasts need to be bias-corrected using the daily annual climatology of station observations
- All other variables need to be bilinearly interpolated to the coordinates of the study-sites.

After the spatial downscaling, seasonal forecast data are interpolated to one-hour temporal resolution with different methods depending on the variable (Terzago et al., in preparation).

The post-processing of seasonal precipitation forecast is shown in this section. The target season is winter, so the 1st of November start date simulations available for the period 1993-2018 for daily precipitation data of SEAS5 are being post-processed. The reference datasets employed are i) ERA5 daily precipitation reanalysis at 0.25° (around 30 km) spatial resolution for the bias correction and for the estimation of the spectral slopes employed in the downscaling procedure and ii) the WorldClim2 monthly climatology at 1 km spatial resolution (Fick and Hijmans, 2017) for generating the precipitation weights used to introduce the orographic effects in the downscaled fields. In this case, the region of study is the Alpine mountain range in central Europe (42-49 °N, 4-11 °E), including the high-elevation stations for which the SNOWPACK model is run (although SEAS5 and ERA5 datasets are locally stored for the global domain).

The RainFARM downscaling method incorporated within CSTools is employed to downscale precipitation which is then used as input for the SNOWPACK model. This method allows taking into account the orographic effects on the precipitation distribution and generates a user-defined number of stochastic downscaling realizations for each member of the original seasonal forecast simulations. For each ensemble member of the seasonal forecast model, we generate 10 stochastic downscaling realizations. In the following subsection, we present the method applied to the SEAS5 model providing 25 ensemble members, such that at the end of the downscaling procedure we obtain a total of 250 fine-scale precipitation fields.

Since the RainFARM downscaling relies on the estimation of the spatial power spectrum of precipitation fields, a squared domain is required. Moreover, this domain has to be larger than the target study area to avoid artifacts/border effects within the target area.

Figure 10 shows the three steps that should be carried out before applying the RainFARM method. Steps 1 to 3 are dedicated to bias correct the precipitation forecast with the quantile mapping correction and compute the spectral slopes and the orographic weights required for the downscaling. In step 4, the downscaling procedure is applied.

| Step 1 | Step 2 | Step 3 | Step 4 |
|---|---|---|---|
| Load daily reference dataset in high resolution | Load daily seasonal forecast and reference dataset (same period and grid) | Get high resolution precipitation climatology | Downscaling with RainFARM |
| Slope | Quantile Mapping correction | RainFARM Weights | Save results |

**Figure 10: Scheme of the steps that need to be carried out to obtain and save a downscaled precipitation dataset. These steps are explained in detail in the text of the manuscript.**

All computations performed in the first step only require the CSTools package. As mentioned above, the spectral slope is calculated using ERA5 at its original resolution and over a larger domain than the target region (37.5-53.25 °N, 2.5-18.25 °E). The path pattern to the data is defined using labels: $STORE_FREQ$, $VAR_NAME$, $YEAR$ and $MONTH$. These labels will be interpreted by *CST_Load*. For instance, the $VAR_NAME$ will be substituted by the information passed by the parameter *var* which in this case is "prlr" that stands for precipitation rate and the $YEAR$ and $MONTH$ will be interpreted from the *CST_Load sdates* parameter which requires a vector of dates in the format "YYYYMM01" where YYYY is the year and MM the month. Then, *CST_Load* retrieves the data from files and arranges it with the following dimensions: *dataset* of length 1 since only ERA5 is being requested, *member* = 1 since this reanalysis only provides one simulation, *sdate* dimension is of length 312 which corresponds to the 26 years of 12 months defined in object "years" with an *ftime* dimension up to 31 corresponding to each day of the month. The remaining dimensions, *lat* and *lon* correspond to the squared domain requested in *CST_Load*. Given that *CST_Load* splits the time series among *sdates* and *ftime* dimension when specifying a forecast dataset, our ERA5 path pattern has been requested through this option. On the other hand, specifying the ERA5 path pattern as an observational dataset (in the *obs* parameter), the function will return a continuous time series from 1993 to 2018 which is less convenient for our purposes here.

In this example over the Alpine domain, the slope of the spatial power spectrum of ERA5 daily precipitation at 0.25° exhibits temporal variability at the seasonal scale. In order to account for this, we calculate the spectral slopes at the monthly time scale, fitting wavenumbers 5 and higher (scales smaller than about 250 km) in order to better reproduce the slope of the spectrum at the small scales (see Terzago et al., 2020 for details). The result of this code are the spectral slopes from January to December.

In the second step, we load the data, taking advantage of library zeallot (Teetor, 2018) that allows us to simplify our code by using an advanced version of the assignment operator (%<-%). Again, the paths to the necessary data must be defined using labels: for the forecast data the path points to the SEAS5 dataset while for the reference data the path points to the ERA5

reanalysis. Thanks to *CST_Load*, these datasets can be interpolated onto a common grid, which, by default, is the grid of the first dataset provided, i.e. the SEAS5 grid. The vector "StartDates" which defines the period of study for November 1st
simulations, is then assigned to the *sdates* parameter. In order to apply the quantile-mapping correction month by month, the function *CST_SplitDim* is used to divide the forecast time dimension in two: one for identifying the days of the month and another to store each month separately.

Step 3 computes the orographic weights from a fine-scale precipitation climatology. In this case, the WorldClim2 dataset
precipitation at 30 seconds resolution is used although other climatologies at high resolution dataset could be used. The WorldClim2 dataset is formatted in tiff files that can be automatically downloaded in the R session thanks to the raster library (Hijmans, 2020). The piece of code for Step 3 shows how to compute the orographic weights for all individual months at once: getting the data from the remote dataset, subsetting for the Alps region with a small increment to correctly compute interpolation (3.5-11.5 °E, 41.5-49.5 °N), and storing the data in an "s2dv_cube" object to be passed to *CST_RFWeights*.

The target resolution is the one most suitable for each specific application. To run the SNOWPACK model, we are interested in the local scale and we choose a target resolution of 0.01°, corresponding to about 1 km. Therefore, the weights and the RainFARM method (step 4) would be computed with a refinement factor (*nf*) of 100. However, such a high refinement factor implies a rather large computational load, and here we show the code using a refinement factor 4. We recommend to the users
to approximate the expected size of the final output, as follows: the original data input of the downscaling step has 25 members, 26 start dates, 31 daily lead times on 8 months covering a region of 8 by 8 grid points, which 8 (bits/byte) times its product is ~ 80 MB; this size will increase by a factor 10 since the realizations and the refinement factor will be applied on both spatial dimensions. For a refinement factor of 100 (4), the expected output is ~80 MB x 10 x 100 x 100 (~ 80 MB x 10 x 4 x 4), so around 8 TB (12.5 GB). The users need to consider that the data size also has implications for the computation time.

Finally, the downscaling method is run in step 4 using the corrected forecast, the slope and weights computed in the previous steps. Note that this code requires high memory resources, although the computation can be split by start date and realization if necessary.

```
# Step 1
library(CSTools)
era5 <- list(name = "era5", path = "/esarchive/recon/ecmwf/era5/$STORE_FREQ$_mean/$VAR_NAME$_f1h-
r1440x721cds/$VAR_NAME$_$YEAR$$MONTH$.nc")
years <- unlist(lapply(1993:2018, function(x){paste0(x, sprintf("%02d",1:12), "01")}))
era5 <- CST_Load(var = "prlr", exp = list(era5), sdates = years, nmember = 1,
                 storefreq = "daily", sampleperiod = 1,
                 latmin = 37.5, latmax = 53.25, lonmin = 2.5, lonmax = 18.25,
                 output = "lonlat")
era5 <- CST_SplitDim(era5, split_dim = "sdate", indices = rep(1:12, 26))
slope <- CST_RFSlope(era5, time_dim = c("sdate", "ftime"), kmin = 5)
```

```
# Step 2
library(zeallot)
StartDates <- paste0(1993:2018, "1101")
exp <- list(name = "ecmwfS5", path =
 "/esarchive/exp/ecmwf/system5c3s/$STORE_FREQ$_mean/$VAR_NAME$_s0-24h/$VAR_NAME$_$START_DATE$.nc")
obs <- list(name = "era5", path =
"esarchive/recon/ecmwf/era5/$STORE_FREQ$_mean/$VAR_NAME$_f1h-r1440x721cds/$VAR_NAME$_$YEAR$$MONTH$.nc")
c(exp, obs) %<-% CST_Load(var = "prlr", exp = list(exp), obs = list(obs),
                          sdates = StartDates, nmember = 25,
                          storefreq = "daily", sampleperiod = 1,
                          latmin = 42, latmax = 49, lonmin = 4, lonmax = 11,
                          output = "lonlat", nprocs = 1)
exp <- CST_SplitDim(exp, split_dim = "ftime")
obs <- CST_SplitDim(obs, split_dim = "ftime")
exp.qm_months <- CST_QuantileMapping(exp, obs, method = "QUANT", wet.day = FALSE,
```

```
                           sample_dims = c("member", "sdate", "ftime"))
exp.qm <- CST_MergeDims(exp.qm_months, merge_dims = c("ftime", "monthly"),
                        na.rm = TRUE, ncores = 4)
```

```
# Step 3
# WorldClim data to s2dv_cube
library(raster)
worldclim <- getData("worldclim", var = "prec", res = 0.5, lon = 5, lat = 45)
wc_month <- lapply(1:12, FUN = function(x) {
                           res <- crop(worldclim[[x]],
                           extent(3.5, 11.5, 41.5, 49.5))
                           res <- as.array(res)
                           names(dim(res)) <- c("lat", "lon", "month")
                           return(res)
                           })
xy <- xyFromCell(crop(worldclim[[1]], extent(3.5, 11.5, 41.5, 49.5)),
                 1:length(crop(worldclim[[1]], extent(3.5, 11.5, 41.5, 49.5))))
lons <- unique(xy[,1])
lats <- unique(xy[,2])
wc_month <- unlist(wc_month)
dim(wc_month) <- c(lat = length(lats), lon = length(lons), month = 12)
wc_month <- s2dv_cube(data = wc_month, lon = lons, lat = lats,
           Datasets = "WorldClim")

weight <- CST_RFWeights(wc_month, lon = exp$lon, lat = exp$lat, nf = 4)
```

```
# Step 4
weights <- Subset(weight$data, along = "monthly", indices = c(11,12,1:1:6))
slope <- Subset(slope, along = "monthly", indices = c(11,12,1:1:6), drop = "non-selected")
fs <- CST_RainFARM(exp.qm, nf = 4,
          weights = weights, slope = slope,
          kmin = 1, nens = 10, verbose = TRUE,
          time_dim = c("member", "ftime"), nprocs = 4,
          drop_realization = TRUE)

fs <- CST_MergeDims(fs, merge_dims = c("ftime", "monthly"), na.rm = TRUE)

fs$Dates[[1]] <- exp$Dates[[1]]
CST_SaveExp(fs, destination = "/esarchive/scratch/nperez/CSTools_manuscript/")
```

Figure 11 shows an example of the results for a random day (the 11th of December 1993) of each post-processing step and the final result for one of the realizations. The monthly spectral slopes obtained in step 1 (Fig. 11c) show an annual cycle. The result of step 2 is a bias-corrected forecast consistent with the reference dataset: the forecast probability density function matches the one for the references, resulting in the same climatology (Appendix C). A simple visual evaluation of the impact of the quantile mapping correction is available in Fig. 11a,b. The results of step 3 (Fig. 11d,f) are the monthly spectral slopes for which values greater (lower) than 1 amplify (reduce) the precipitation signal from the seasonal forecast. Finally, the spatial resolution improvement given by RainFARM for a specific date when applying a refinement factor 4 or 100 is shown in Fig. 11e,g.

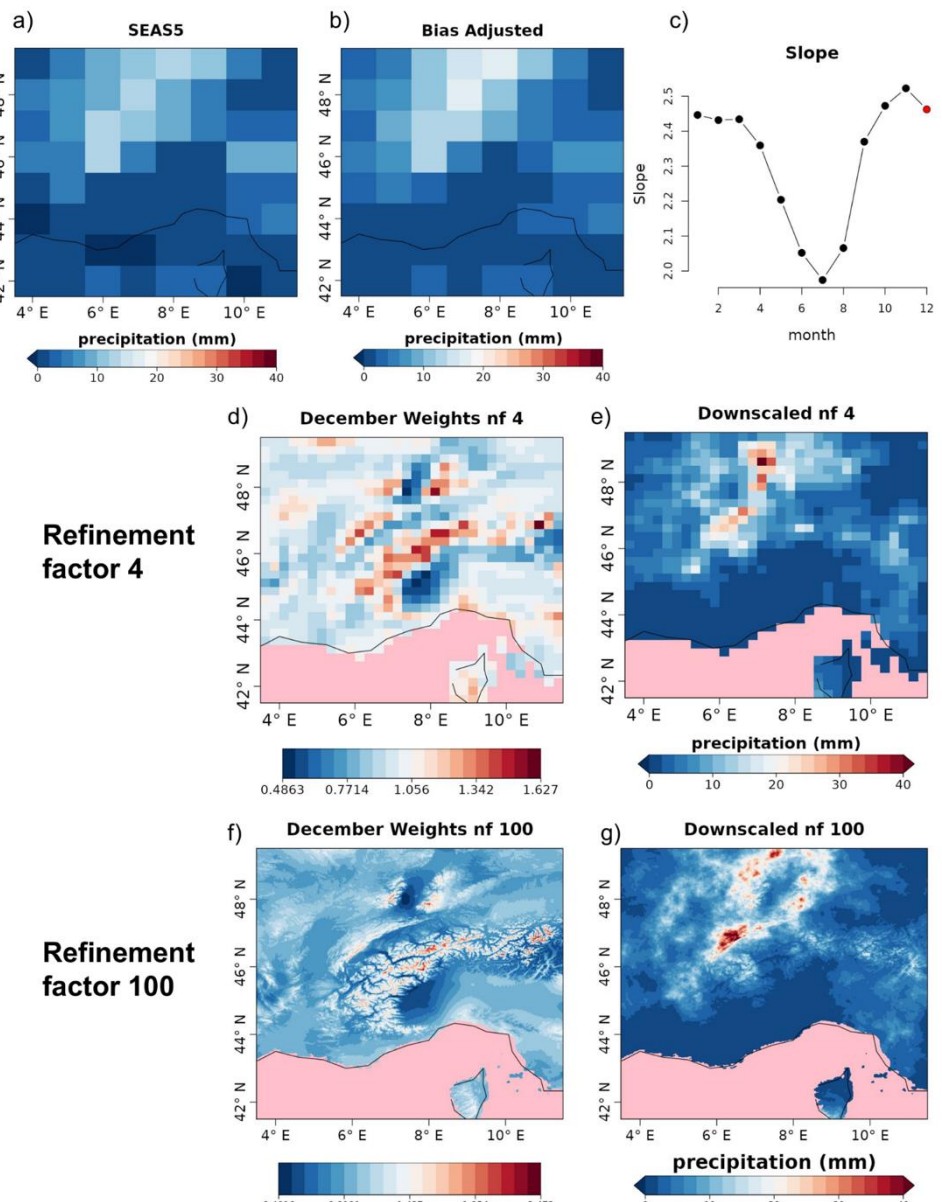

**Figure 11: One out of 25 ensemble members of the original ensemble (a) large-scale precipitation field for SEAS5 for the 11th of December 1993 to be downscaled; (b) bias-corrected field; (c) monthly slopes (the slope used for downscaling is highlighted in red). In the middle (refinement factor 4) and bottom (refinement factor 100) rows the comparison of the weights (d and f) and the downscaled (e and g) precipitation fields for SEAS5 for the 11th of December 1993 are shown. Grid points for which no data are available are coloured in pink.**

770 The SNOWPACK model is run with each of the 21 seasonal forecasts using 7-month forecasts initialized on November 1st over the hindcast period 1996-2016 in order to simulate the most relevant period for the snow dynamics, i.e. November-May. Figure 7 shows an example of the SNOWPACK model output, and specifically, the snow depth forecasts obtained from the SEAS5 forecast initialized on the 1st of November 2006, for the area including the station of Bocchetta delle Pisse —2410 m above sea level. For each forecast, we obtain an ensemble of 250 snow depth /SWE simulations, derived from ten downscaling

775 realizations for each of the 25 precipitation ensemble members of the SEAS5 forecast system. SEAS5-SNOWPACK forecasts are able to reproduce the variability of the observed snow depth (Fig. 12). For 2006/2007 in particular, the median forecast is lower than the model climatology, which is consistent with the low amount of snow that was observed that winter.

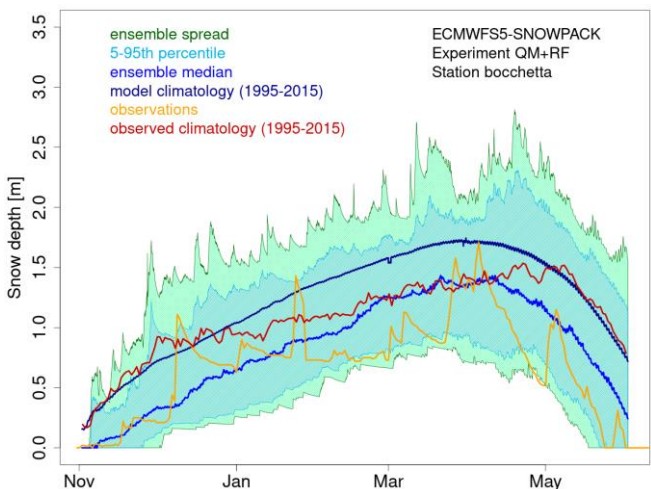

**Figure 12: Seasonal forecast of snow depth obtained from the SNOWPACK model driven by the SEAS5 seasonal forecast system data. The forecast, initialized on the 1st of November 2006 and covering the 7 following months, refers to the station of Bocchetta delle Pisse, 2410 m a.s.l. in the North-Western Italian Alps. The green shadow shows the spread of the 250 daily snow depth forecast ensemble members; the cyan shadow represents the 5th-95th percentile range of the forecast distribution; the blue line represents the ensemble median of the 250 ensemble members for the 2006/2007 season; the dark blue line represents the model climatology (mean over the seasons 1995-2015 and over all ensemble members); the red line represents the observed climatology and the orange line represents the station observations for the 2006/2007 season.**

### 3.3 Use case 3: Seasonal forecasts for a river flow

In this last use case, we provide downscaled and bias-adjusted seasonal forecasts of daily maximum, minimum and mean temperature and precipitation. This use case can be relevant for instance to climate scientist, hydrologists and developers of agriculture services because these variables are often used to calculate the evapotranspiration using the Hargreaves and Samani equation (e.g. Oudin et al., 2005). Indeed, while the typical seasonal forecasts are provided at a resolution of around 100 km, the spatial resolution required by some hydrological model is 5 km (Roulin and Vannitsem, 2005). Rather than giving detailed code instructions similar to the previous use cases, an extensive discussion on the data-generation process is provided and the code is made available online. Note again that all steps are being executed using only CSTools functions.

The domain in this case covers the Aliakmon basin and a large part of Greece. Apart from the variables aimed to post-process, the sea level pressure (SLP) is used as a requirement for the downscaling method chosen. As seasonal forecast data, all the variables are taken for the 25 members of SEAS5 initialized on May 1$^{st}$ for the 1993-2019 period. For the reference rainfall dataset, the daily data from CHIRPS (Climate Hazards group Infrared Precipitation with Stations; https://data.chc.ucsb.edu/products/CHIRPS-2.0/; Funk et al. 2015) is taken, available at 0.05° resolution is used. Moreover, this dataset incorporates corrections for different mountain elevations and slopes, which is relevant in the orographically complex Aliakmon basin with elevations above 2000 m. For temperature and SLP, we use the ERA5-Land (Muñoz-Sabater et al., 2021) available at around 0.1° and further downscaled to 0.05° resolution using a simple lapse-rate correction.

An analog approach is used, which combines both the synoptic-scale pressure (through the exploration of SLP) over Europe and the regional-scale rainfall over Greece. For the best analog day, the high-resolution fields of both temperature and rainfall over Greece are considered as the end product. This approach ensures the spatio-temporal consistency of both fields.

Figure 13 provides the step-by-step structure of the methodology used while Fig. 14 provides a visual representation of the post-processing chain. As shown in Fig. 13, the overall methodology can be separated into a bias adjustment phase (steps 1-2) and a downscaling phase (steps 3-5) that uses an analog approach. The bias adjustment phase starts by loading (using *CST_Load*) the daily forecast and observational rainfall data over Greece upscaled at 1° resolution. In step 2,

these daily rainfall forecasts are bias-adjusted (using *CST_Calibration* with method *bias*) against the CHIRPS dataset. The bias adjustment is done per month of lead time using a leave-one-out or cross-validation approach. Breaking up the data per month was done using the *CST_SplitDim* function.

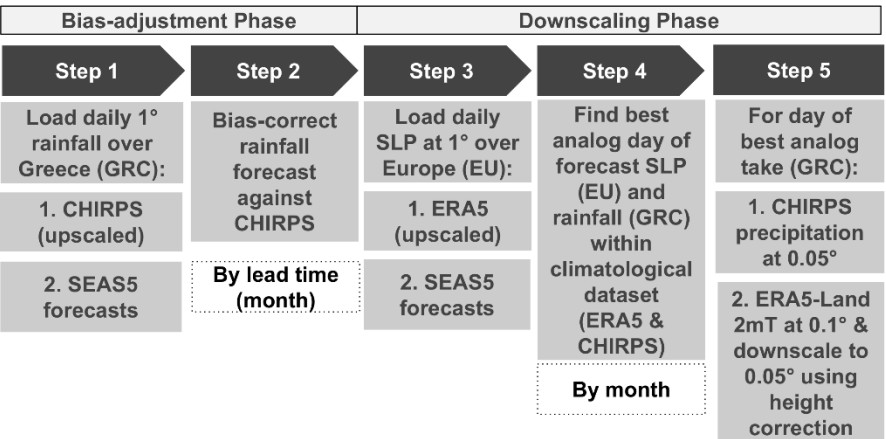

**Figure 13: Scheme of the necessary steps to obtain bias adjusted and downscaled input. The abbreviations used are SLP, SEAS5, EU (Europe) and GRC (Greece).**

The downscaling phase starts in step 3 (see Fig. 13) by loading the sea-level pressure (SLP) of the forecast and reference at 1° resolution over Europe. In step 4, both the bias-adjusted precipitation fields of a particular forecast day over Greece and the SLP anomaly fields over Europe of a particular forecast day are selected. These fields are then compared to all the fields of a large climatological reference dataset in order to find the best analog (using the *Analogs* function). This dataset covers the period 1993-2019 but includes only days with the same month as the selected day (and excludes the selected day). Separating the data per month was again done using the *CST_SplitDim* function. The criterion to find the best analog is called *Local_dist* and minimizes the Euclidean distances of the large-scale SLP and the local-scale rainfall patterns, both at 1° resolution. Finally, for the day corresponding to the best analog, the CHIRPS precipitation field at 0.05° resolution is then selected as the bias-adjusted and downscaled field of the selected day (See Fig. 14c and 14d). In order to obtain the temperature, the ERA5-Land dataset over Greece is considered for the day of the best analog. More specifically, the ERA5-Land daily minimum, maximum and average temperature at 0.1° resolution are downscaled to a resolution of 0.05° using lapse-rate height corrections. Finally, the downscaling procedure steps 4-5 are iterated over each day of every ensemble member of the seasonal forecast in order to obtain a fully bias-corrected and downscaled seasonal forecast over Greece.

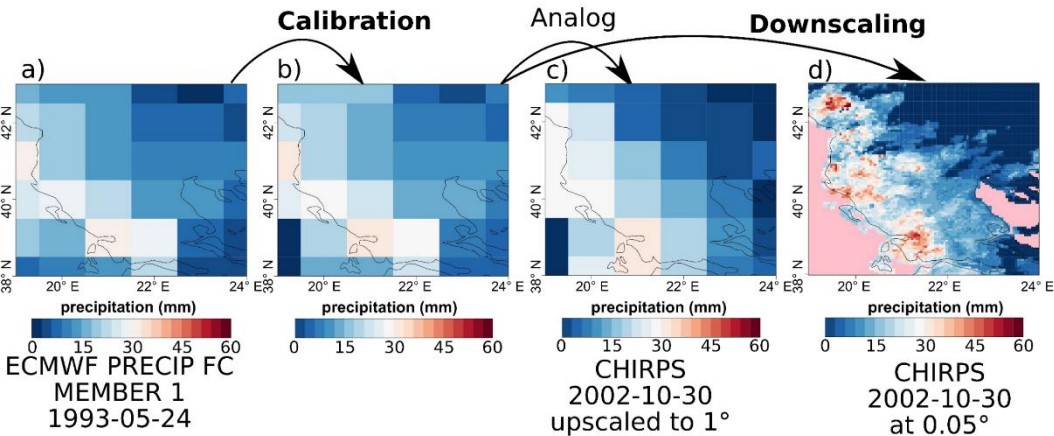

**Figure 14: Comparison of the original forecast (a), bias-adjusted forecast (b), the analog (c) and the downscaled (d) forecast over Greece for SEAS5 for the 24th of May 1993.**

**4 Conclusions**

CSTools contains state-of-the-art methods to post-process seasonal forecast datasets specially focusing on statistical correction
and downscaling methods, as well as classification methods. These methods are extremely valued in the community given the need of correcting intrinsic systematic model errors and the need of many final applications to have these forecasts in higher resolution than the original resolution provided by the forecast systems. On the other hand, the visualization tools tailored for probabilistic forecasts are able to summarize the results in a concise yet user-friendly manner.

Three use cases showcased the ability of the CSTools R package to comprehensively post-process seasonal forecasts in the context of scientifically advanced impact modelling. The final users potentially interested in these three use cases represent a classical current-day sample of the users (and disciplines) that can benefit from CSTools. The energy sector can see the utility of seasonal forecast post-processed with CSTools in all the use cases presented: the first one showed the potential of seasonal forecasts to anticipate high wind speed events in the Iberian peninsula and the impact it had on energy production and prices;
the second and third cases could be of high interest for the hydrological energy sector since foreseeing months in advance the snow depths at high altitudes and the stream flow in catchments may allow hydropower managers to plan their activity. Similarly, these use cases are relevant for risk management of high wind speed, coastal and flooding events, as well as, agricultural issues implied by droughts, irrigation needs or water resources management.

In addition to the climate post-processing methods, two aspects of the CSTools design are highly valuable: the data loading feature allows users to easily load and arrange the forecast and the reference datasets in a common structure (i.e. the "s2dv_cube" object) simplifying the subsequent data manipulation steps, and the internal use of the multiApply package in the data processing functions makes them flexible to work with any number of dimensions and allows parallel computation. Finally, its compatibility with startR package allows to process datasets larger than the available RAM memory.

The development guidelines are a fundamental piece of documentation for the future extension of the package when new state-of-the-art methods are required or become available. These guidelines are already being adopted by another R package called CSIndicators that stands for Climate Services Indicators, dedicated to calculating the most suitable tailored indicator for each particular climate service application (agriculture, food security, energy, water management…). Other types of documentation,
such as vignettes, provided along with the package, are intended to facilitate the users' learning process.

**Appendix A: Details on data collection, curation, homogenization, and requirements for CST_Load**

In order to use *CST_Load*, the storage needs to be homogenized. *CST_Load* accepts several parameters to configure the loading and interpolation of data. The *CST_Load* documentation in the reference manual is linked to the s2dverification (Manubens et

al., 2018) reference manual where the description of all parameters is detailed.

Basically, *CST_Load* requires path patterns pointing to the NetCDF files or OPeNDAP URLs requested via the other parameters. A variable with a matching name must be present in the files. The path patterns, one for each experimental/observational dataset to be loaded, express the set of files comprising each dataset. Therefore, a path pattern is a

string containing some specific wildcards that are recognised and replaced by the corresponding values by *CST_Load*. The most commonly used wildcards in a path pattern specification are "$START_DATE$", "$STORE_FREQ$", and "$VAR_NAME$". For example, given a dataset that consists of the following files:

- /data/datasetA/monthly/tas_20180101.nc
- /data/datasetA/monthly/tas_20180201.nc

- /data/datasetA/monthly/tas_20180301.nc

The path pattern to express the set of files would be as follows:

"/data/datasetA/$STORE_FREQ$/$VAR_NAME$_$START_DATE$.nc"

The use case 1, "Assessing the odds of an extreme event", directly loads wind speed (in surface for the case of SEAS5 and at 100m for the case of ERA5). However, this variable is not directly available in the Copernicus CDS while u and v wind components are in 6 hourly and monthly frequencies. In order to get the monthly wind speed, the 6 hourly frequency components are used to calculate the 6 hourly wind speed and then, calculate the monthly average of the wind speed using CDO (Schulzweida, 2019). Notice that averaging the monthly wind components may lead to a different result. To automatise

this calculation on all the files in a folder, the following bash code could be adapted:

```
path_output="./data/monthly/sfcWind_f6h/"
 path_component="./data/6hourly/"
for year in {1993..2018}
do
for month in 01 02 12
do
output_file=${path_output}"sfcWind_${year}${month}01.nc"
uas_file=${path_component}"uas/uas_${year}${month}01.nc"
vas_file=${path_component}"vas/vas_${year}${month}01.nc"
cdo -L -b F64 -f nc -setunit,'m/s' \
  -setname,"sfcWind" \
  -sqrt \
   -add \
    -mul -selname,uas $uas_file -selname,uas $uas_file \
    -mul -selname,vas $vas_file -selname,vas $vas_file \
  ${output_file}_tmp
cdo monmean ${output_file}_tmp $output_file
done
done
```

An equivalent script, using CDO *dailymean* operator, can be used to convert the required variables in use case 2 and 3 into daily mean values after datasets are downloaded.

 **Appendix B: CSTools functions synthetic description**

**Table A1. Summary of the functions and methods by category including a description, as well as the origin of the first known code and the references. Prefix "CST_" refers to functions working on a specific object class called "s2dv_cube" while those without the prefix, accept multi-dimensional arrays with named dimensions as input. Asterisks indicates functions that are used in vignettes.**

| Category | Function | Method description | Original development? | Reference |
|---|---|---|---|---|
| **Retrieval and transformation** | CST_Load | Retrieves experiment and reference data from files stored in a common format. Includes regriding options. | Adaptation to CSTools | Manubens et al., 2018 |
| | CST_Anomaly* | Calculates anomalies from experiment and reference data with or without cross-validation. | Adaptation to CSTools | Manubens et al., 2018 |
| | CST_SaveExp | Saves experimental data (with ensemble dimension) into NetCDF files (one for each start date). | Yes | |
| | CST_MergeDims | Transforms the data array with named dimension by merging two requested dimensions. | Yes | |
| | CST_SplitDim | Transforms the data array with named dimensions by splitting a requested dimension following a user-defined frequency or pattern. | Yes | |
| | as.s2dv_cube | Converts data loaded using the startR package or s2dverification Load function into a 's2dv_cube' object. | Yes | |
| | s2dv_cube | Returns a s2dv_cube object by providing the data and metadata through its arguments. | Yes | |
| **Classification** | CST_MultiEOF | Applies an EOF analysis over multiple variables retaining the minimum number of principal components needed to reach the user-defined variance. | Yes | |

| | CST_WeatherRegimes* | Applies a cluster analysis based on the user-defined number of clusters. A PCA could be requested to reduce the dimensionality of the dataset. | Yes | Cortesi et al., 2019; Torralba, 2021 |
|---|---|---|---|---|
| | CST_RegimesAssign* | Matches patterns with a set of reference maps (i.e. clusters from CST_WeatherRegimes) based on the minimum Eucledian distance or the highest spatial correlation. | Yes | Cortesi et al., 2019; Torralba, 2021 |
| | CST_CategoricalEnsCombination | Converts a multi-model ensemble forecast into a categorical forecast by giving the probability for each category. Different methods are available to combine the different ensemble forecasting models into probabilistic categorical forecasts: | Yes | |
| | | "pool" for ensemble pooling where all ensemble members of all forecast systems are weighted equally; | | DelSole et al., 2013 |
| | | "comb" for a model combination where each forecast system is weighted equally; | | DelSole et al., 2013 |
| | | "mmw" for model weighting. | | Rajagopalan et al. 2002; Robertson et al. 2004; Van Schaeybroeck and Vannitsem, 2019 |

| | | | | |
|---|---|---|---|---|
| | CST_EnsClustering* | Groups ensemble members according to similar characteristics and selects the most representative member for each cluster. The user chooses which feature of the data is used to group the ensemble members: time mean, maximum, a certain percentile (e.g. 75 standard deviation) or trend over the time period. | Adaptation to CSTools | Straus et al., 2007 |
| **Downscaling** | CST_Analogs* | Searches for days with similar large-scale conditions (i.e. analogs) to provide downscaled fields. | Yes | Yiou et al, 2013 |
| | CST_RainFarm* | Implements the Rainfall Filtered Autoregressive Model which is a stochastic downscaling procedure based on the nonlinear transformation of a linearly correlated stochastic field. | Adaptation to CSTools | Rebora et al. 2006a,b; D'Onofrio et al. 2014; Terzago et al. 2018 |
| | CST_RFTemp | Downscales a temperature field by using a simple lapse rate correction. | Yes | |
| | CST_AdamontAnalog | Identifies analog fields in a reference dataset, based on corresponding weather types (requires CST_AdamontQQcor beforehand) | Adaptation to CSTools | Verfaillie et al., 2017 |
| | CST_AnalogsPredictors | Downscales precipitation and maximum and minimum temperature using analogs and considering synoptic situations and significant predictors | Adaptation to CSTools | Peral García et al., 2017 |

| | | | | |
|---|---|---|---|---|
| | CST_BEI_Weighting* | Returns a weighted ensemble mean (or weighted terciles probabilities) according to the skill of individual members at predicting a climatological index (e.g.: NAO) (requires BEI_PDFBest and CST_BEI_Weighting beforehand). | Yes | Sánchez-García et al. 2019 |
| **Correction** | CST_Calibration | Member-by-member bias correction. Different methodologies are available. | Yes | |
| | | "bias" corrects only the mean bias. | | Torralba et al. 2017 |
| | | "evmos" applies a variance inflation technique to ensure the correction of the bias and the correspondence of the variance between forecast and observation. | | Van Schaeybroeck and Vannitsem, 2011 |
| | | "mse_min" corrects the bias, the overall forecast variance and the ensemble spread by minimizing a constrained mean-squared error. | | Doblas-Reyes et al. 2005 and Torralba et al., 2017 |
| | | "crps_min" corrects the bias, the overall forecast variance and the ensemble spread and minimizing the Continuous Ranked Probability Score (CRPS). | | Van Schaeybroeck and Vannitsem, 2015 |
| | | "rpc-based" adjusts the forecast variance, ensuring that the ratio of predictable components (RPC) is equal to one. | | Eade et al. 2014 |
| | CST_QuantileMapping | Quantile mapping adjustment for daily (or sub-daily) data. | Adaptation to CSTools | Gudmundsson et al., 2012; Gudmundsson, 2016 |

| | | | | |
|---|---|---|---|---|
| | CST_DynBiasCorrection | Applies a bias correction between the model and the observations using the division into terciles of the local dimension 'dim' or inverse of the persistence 'theta'. Model values with lower 'dim' will be corrected with observed values with lower 'dim', and similarly for theta (requires Predictability and CST_ProxiesAttractor beforehand). | Yes | Faranda et al., 2017; Faranda et al., 2019 |
| **Verification** | CST_MultiMetric* | Computes correlation, root mean square error and the root mean square error skill score for individual models and multi-model mean. | Adaptation to CSTools | Manubens et al. 2018. Mishra et al., 2019 |
| | CST_MultivarRMSE* | Calculates the RMSE using multiple variables simultaneously. | Yes | |
| **Visualization** | PlotCombinedMap* | Plots multiple lon-lat variables in a single map according to a decision function. | Yes | Mishra et al., 2019 |
| | PlotForecastPDF* | Plots the probability distribution function of several ensemble forecasts. Can include tercile and extreme (above P90 and below P10) categories, individual members and a corresponding observation. | Yes | Soret et al., 2019; Lledó et al., 2020a |
| | PlotMostLikelyQuantileMap* | Plots the probability for the category with the maximum probability in each grid point. | Yes | Lledó et al., 2020a; Torralba, 2019 |
| | PlotPDFsOLE | Plots two probability density gaussian functions and the optimal linear estimation (OLE) resulting from their combination. | Yes | Sánchez-García et al., 2019 |

| | PlotTriangles4Categories* | Function to convert any 3-d numerical array to a grid of coloured triangles. | Yes | Torralba, 2019; Verfaillie et al., 2020; Lledó et al., 2020b |
|---|---|---|---|---|

**Appendix C: Probabilities distribution use case 2**

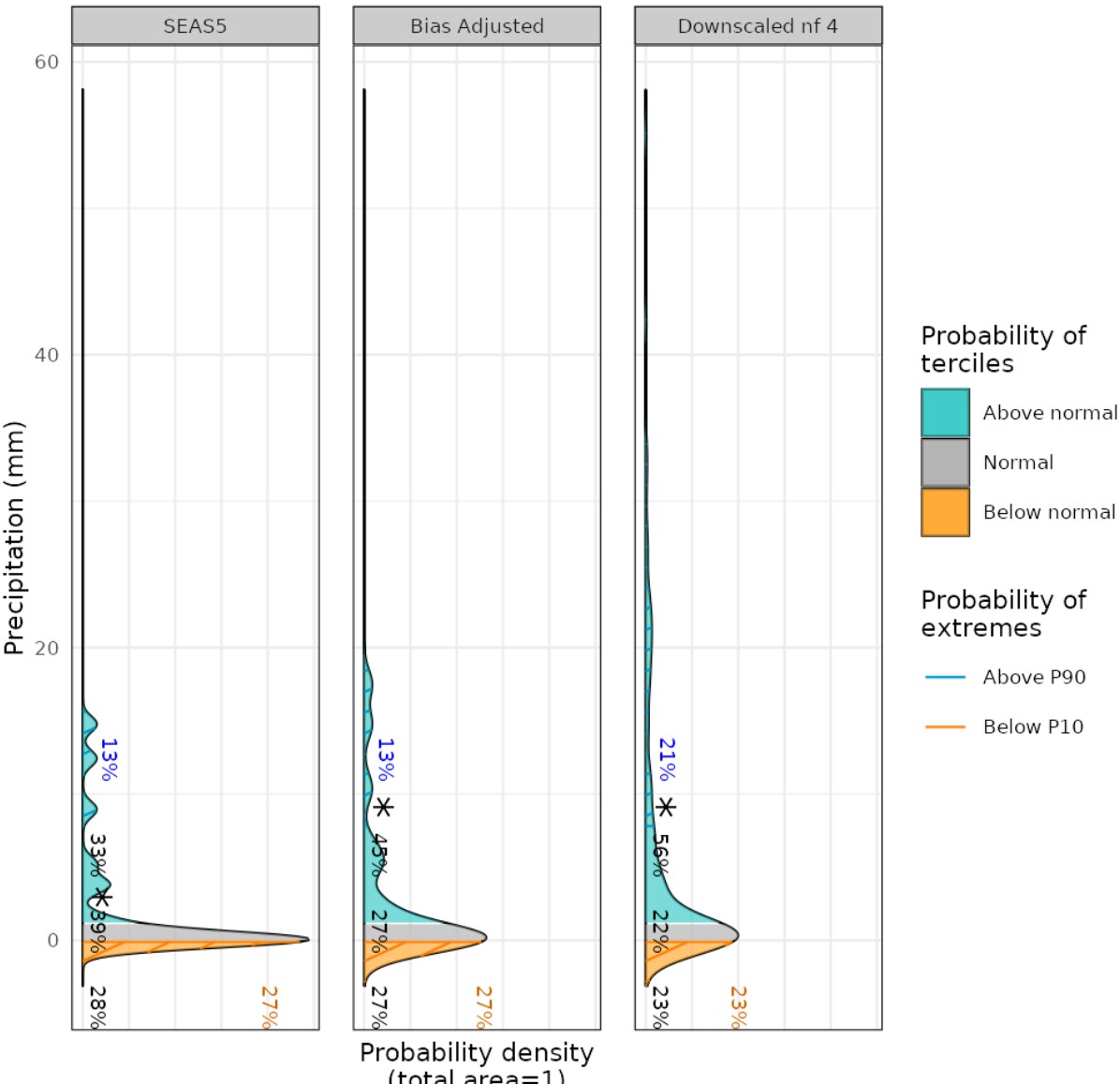

**Figure A1: From left to right, PDF for the original, the bias-corrected by quantile mapping and the downscaled precipitation data (for refinement factor 4) SEAS5 for a grid point corresponding to a observational site in the Alps on the 11th of December 1993. For each PDF, three categories of equal size are shown: terciles above normal (blue), normal (grey) and below normal (orange), defined according to the area average of ERA5 reanalysis for the period 1993-2018. Percentages represent the forecast probabilities of each tercile, the most likely tercile is highlighted with a star and the blue and orange percentages represent the probabilities for P10 and P90 (hatched areas), respectively.**

## Code availability

CSTools is released under the Apache License version 2.0. The latest release of CSTools 4.0.1 is publicly available on CRAN repository https://CRAN.R-project.org/package=CSTools, it is being developed at BSC-CNS GitLab repository https://earth.bsc.es/gitlab/external/cstools/ and shared in the Zenodo DOI:10.5281/zenodo.5549474. The code to reproduce the use cases and plots shown in this manuscript is shared in the three sites and we recommend to find it at the GitLab repository https://earth.bsc.es/gitlab/external/cstools/-/tree/master/inst/doc.

## Author contribution

Núria Pérez-Zanón developed several functions in the package: *CST_SaveExp*, *CST_SplitDims*, *CST_MergeDims*, *CST_QuantileMapping*, *CST_MultiMetric*, *s2dv_cube* and *as.s2dv_cube*. She, as maintainer, also co-managed the package with Louis-Philipe Caron. Silvia Terzago and Bert Van Schaeybroeck together with Emmanuel Roulin designed the second and third use cases presented in the manuscript, respectively. Llorenç Lledó created the function *PlotForecastPDF* and designed the first use case presented in this manuscript. Nicolau Manubens provided advice on the design of the API and compatibility with other packages, drafted the CSTools development guidelines, developed the *CST_Load* and *PlotCombinedMap* functions and created the sample data provided along with the package. Carmen Alvarez-Castro created the *Analogs* function and the dynamical bias correction methodology. Lauriane Batté adapted the ADAMONT downscaling methodology to CSTools. Bert Van Schaeybroeck also developed the *CategoricalEnsCombination* function and the methods bias, evmos, and crps in *Calibration* function. Verónica Torralba developed the mse_min method while Carlos Delgado-Torres added the rpc-based method to the *Calibration* function. Pierre-Antoine Bretonnière created the cds-seasonal-downloader code. Susana Corti developed the original code of *EnsClustering*. Marta Domínguez adapted the *AnalogsPredictors* downscaling methodology to be included in CSTools. Federico Fabiano and Ignatzio Giuntoli developed the code of *MultiEOFs* and *EnsClustering*. Jost von Hardenberg developed the RainFARM functionalities, as well as the *RFTemp*. Eroteida Sánchez-García coded the BEI methodology and the PlotPDFsOLE visualization function. Verónica Torralba also coded the *BiasCorrection*, *WeatherRegime*s, *RegimesAssign*, *PlotMostLikelyQuantileMaps* and *PlotTriangles4Categories* functions. Deborah Verfaillie created the *CST_MultivarRMSE* function. Núria Pérez-Zanón prepared the manuscript with contributions from all co-authors.

*Competing interests*. The authors declare that they have no conflict of interest.

## Acknowledgments

This work has been performed in the framework of the MEDSCOPE (MEDiterranean Services Chain based On climate PrEdictions) ERA4CS project (GA 690462) with contributions from the Sub-seasonal to Seasonal climate forecasting for Energy (S2S4E; GA 776787) and the European Climate Prediction system (EUCP; GA 776613) from the European Union's Horizon 2020 research and innovation programme. CDT acknowledges the funding by the Spanish Ministry for Science and Innovation (FPI PRE2019-088646). We thank other contributors to the CSTools package: Filippo Calì Quaglia, Chihchung Chou, Nicola Cortesi, Paolo Davini, Raül Marcos, Niti Mishra, Jesus Peña, Francesc Roura-Adserias and Danila Volpi. We also thank Margarida Samso for their support on data formatting, An-Chi Ho for her work on s2dv, s2dverification and startR packages, and Kim Serradell and Francesco Benincasa for the technical support. The R community and especially the developers of the packages rainfarmr, multiApply, ClimProjDiags, ncdf4, plyr, abind, data.table, reshape2, ggplot2, qmap,

RColorBrewer, verification, zeallot, testthat, knitr, markdown and rmarkdown are sincerely acknowledged, as well as the CDO

developers.

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
