# Peer review of "The CSTools (v4.0) Toolbox: from Climate Forecasts to Climate Forecast Information"

_Geoscientific Model Development, 2021_

## Author Comment (AC1)

**Answer to RC1: 'Comment on gmd-2021-368', Anonymous Referee #1, 04 Jan 2022**

> For example, the authors acknowledge that climate data retrieval/loading and formatting (of the NetCDF files prior the import in R or python environments) is often a blocking point for some users: "This can be a labour-intensive step when trying to combine multiple datasets such as observations and forecasts from multiple systems" (l.68). However, the current manuscript simply refers to external notebooks without further comments (l.196-203). In general, there are file paths, in both this paper and vignettes (GitHub), pointing where data is stored (apparently locally) with no reference or link to retrieve the original input files. Following the link Earth Sciences / CDS Seasonal Downloader · GitLab (bsc.es), leads the reader to python functions such as "download_seasonal_cds_monthly.py", which raises the question of the consistency of the tool in terms of programming language (R, CDO, python).

We would like to clarify that the aim of the software toolbox is to provide post-processing methods (i.e.: correction methods for forecast calibration, classification methods for multi-model forecast combination or scenario selection, downscaling methods, and visualization tools) that aren't currently available in other software packages. We have included some existing methods to facilitate the comparison of the results by gathering all these methods into a single software toolbox.

We are aware that additional steps are required in the post-processing chain, namely data retrieval and formatting. In order to facilitate the use of the toolbox, we offer additional tools. These tools are used to download data from the C3S climate data store and were designed in python. The users are free to use them or not.

Furthermore, although the Copernicus Climate Data Store is one of the main sources of climate datasets, there exist other data repositories (e.g.: National Center for Environmental Information NCEI). The different data repositories can deliver the datasets in different formats for both file and structure, making it challenging to create a single software/function that considers all requirements.

As mentioned in line 205, we have also considered the possibility that users retrieve the data (from local or remote data storage) by other means than the function provided in CSTools (CST_Load). Other packages, such as ncdf4, raster or ecmwfr, already exist to download or retrieve data from files. Therefore, users could have their own code to cover this need. If that is not the case, we provide a python code to download and format datasets from the CDS to comply with CST_Load requirements. Once the data is loaded in the R session, the user should understand that the most basic data types accepted by CSTools are arrays with named dimensions, which is explained in this manuscript and in the package documentation.

In section 1.2, we have added a new first step to the list of steps to explain the climate forecast post-processing chain:

• Data collection, curation and homogenization: This includes collection of data from heterogeneous remote data sources, storage and indexing into local or organisation-accessible file systems or servers, and homogenization for all data files to comply with common internal conventions. The complexity of this step can be high particularly if the data sources do not follow community standards. This step is out of the scope of this manuscript and the CSTools toolbox, and the use of other tools such as the cds-data-downloader (https://earth.bsc.es/gitlab/es/cds-seasonal-downloader) is suggested for this purpose.

In the same section, we also clarify:

The primary aim of CSTools is, therefore, to share post-processing methods (i.e. correction methods for forecast calibration, classification methods for multi-model forecast combination or scenario selection, downscaling methods, and visualization tools) that aren't currently available in other software packages or, whether the method exists in separate software, their inclusion facilitates the comparison of the results. Because additional steps are required (i.e. data retrieval from remote servers, storage and, indexing into local or organization-accessible file systems or servers, curation and formatting, and finally loading from the file systems or servers onto RAM memory of the processing machines), we provide extra functions and scripts in order to facilitate the use of the toolbox.

The datasets used in the use cases are cited in the text for reproducibility and a new appendix is included to give details on how we have created the local storage.

Finally, CDO is used internally by the CST_Load function to interpolate the data when the user requires it, for instance, when comparing datasets that are not on the same grid. However, the reliance on CDO remains invisible to the user. CDO is widely used for this purpose in the climate community, so we deemed it was unnecessary to develop new code, provided it could be integrated into the R framework.

> *This manuscript is not self-contained enough to allow a quick grasp of the R functions while having a precise idea of the underlying methods.*

Thank you for this comment. Since CSTools aggregates a wide range of state-of-the art methods, balancing the general information with specific details is difficult. We attempted to provide sufficient background information by adding a short discussion for each method together with literature references for readers interested in learning more. Considering this comment and some other comments below, we updated Table 1 to summarize the functions and methods in CSTools.

[revised manuscript text omitted]

**Some minor adjustments could help broaden the target audience of the paper.** For example, make sure each specific acronym or term is introduced. This toolbox aims at facilitating the integration of climate data in "*sectoral applications*", yet the manuscript is hardly accessible to potential interested parties, who would not necessarily have the knowledge of all the techniques recently developed in the field of climate services. Although it is acknowledged that the paper aims at experts (*"applied climate scientists or climate services developers*" l.59), in practice, specialists who already handle climate data frequently have certainly developed their own routines and procedures to perform most of the stated operations, so extending the target audience to non-experts could truly add value to this manuscript. Indeed, especially in the context marked by an increasing concern about climate change, this toolbox would gain from being more understandable by energy system planners (l.530) but also mathematicians, risk modelers, insurers, economists, agricultural engineer, etc. Currently, although the manuscript includes ample references, authors often use specific acronyms or terminology (in particular in 2.2) without proper and non-technical introduction/definitions

> that would allow the target audience to be broadened (e.g. "Best Estimate Index" l.301, "ignorance score" l.250, "SEAS5" l.430, etc.).

We appreciate this comment which will allow the manuscript to be more readable and accessible to a wider audience. We have revised the manuscript and added clarifications throughout the document. For example:

l.301 The BEI is an acronym that was casted to identify the method described in Sánchez-García et al. 2019. For that reason, we have tried to rewrite the paragraph:

Previous version: Best Estimate Index (BEI) is a methodology that can be used to improve the forecast skill when a relationship exists between a climatological index and a given climate variable as shown in Sánchez-García et al. (2019), where the technique is shown to improve the skill for precipitation over the Iberian Peninsula using the North Atlantic Oscillation (NAO).

New version: Sánchez-García et al. (2019) used the North Atlantic Oscillation (NAO) to improve the skill of the seasonal precipitation forecast over the Iberian Peninsula. Given that this methodology could be explored to improve the skill of different climate variables that are led by other climate indices in different regions, the method has been generalized in CSTools and named Best Estimate Index (BEI).

l.250 More specifically, this method uses different weights for the occurrence probability predicted by the available models and by a climatological forecast and optimizes the weights by minimizing the ignorance score, which is a measure of the information conveyed by a forecast (Tödter and Ahrens, 2012).

l.430

Previous version: the ECMWF SEAS5 system, obtained from C3S (SEAS5)

New version: the latest ECMWF long-range forecasting system SEAS5, obtained from C3S (SEAS5; Johnson et al. 2019)

l.727 the acronym for above sea level (a.s.l.) has been removed.

> **The overview (section 2) should add value to R package and documentation with further abstraction and description of the underlying processes.** For example, the underlying procedures are not described mathematically in the current version of the manuscript. We could expect this paper to better describe and focus on the "processes" (mathematical specification, parameters, hypothesis), especially as the functions and attributes are already described in the R documentation and vignettes.

We understand this concern but we consider that this granularity of description would lead to an excessive increase in the length of the manuscript while it would be a duplication of the references provided. We consider that the revised Table 1 will simplify the search for references by allowing users to dig into the detailed mathematical

description of each method. Furthermore, the software is open so the user can see the code in case he/she would like to learn more about the calculations.

> **Regarding the presentation, the authors too often use bullet points (listing).** This manuscript sometimes looks more like a complementary user guide made of "lists" than a model description paper. This is particularly true for section 2. This problem of structure affects the substance because even if each of the functions is described in an understandable way, a linear reading of the manuscript makes it difficult for the reader to retain the main mechanisms and methodological choices the package embeds. The structure of the use cases (section 3) should be streamlined to facilitate the reading, e.g. (i) application (why?), (ii) data required/ input and at what resolution / frequency, (iii) process required from source to model, (iv) code guide, (v) output and final data visualization and (vi) interpretation. Some sentences/paragraphs which refer to documentation or with links, could be removed or placed in footnotes. In addition, there are some (not always working) links in the text while we would rather have the information in the document, whereas there are code boxes with path to NetCDF files without the link of website to retrieve the data.

As aforementioned, balancing the provision of in-depth information with a good narrative was a difficult task. Therefore, we really appreciate the constructive suggestions.

We have modified the structure of some sections to improve the readability of the manuscript. The bullet points in section 2.1, regarding the benefits of the nested structure of the functions, have been removed. However, we feel that the current format of Section 2.2 provides the reader with a quick reference for any given function, which would otherwise be lost if the style was changed. Finally, we have re-structured section 3 following the scheme suggested and removed non-working links. The data repositories and references for the datasets are mentioned in the text we have also included an appendix explaining the repository homogenization needs.

> **Contributions.** If one of the contributions is the "gathering" of existing functions in a harmonized toolbox, it is hard to say if some of the processes are original or not in the current version.

To solve this issue, we have reformatted Table 1 including now the literature where the novelty is outlined. It can be seen that most references are in recent, high-level journals. Table 1 also shows the functions that were originally coded for CSTools. Other functions, like CST_RainFARM, used existing software and were fitted to meet the requirements of CSTools. Hopefully, Table 1 makes this clearer now.

> **Paper hardly self-contained.** In general, the paper requires to know or check references and nothing can be done from scratch based on the description given in this manuscript only.

We understand the effort that is required to explore new software and methods and we consider the manuscript a starting point for new users interested in using the software.

They can find details on the reference (following the reformatted table 1) they would need to check depending on the methods they would apply. However, describing in detail each of the functions would make the manuscript prohibitively long. Finally, the use cases included in the manuscript can be used as a starting point and adapted to the reader's specific problem, thus easing the use of the package.

> **Conclusion and recommendations:** The paper does not present a model but a toolbox to introduce climate data in several applications. This toolbox fills a clearly identified gap and could help researchers addressing relevant scientific questions within the scope of EGU. This paper proposes no substantial advance I could identify, but from an operational standpoint, the proposed package is within the scope of GMD and the amount and quality of supplementary material is significant. However, in the current manuscript I would recommend the authors to provide more information about (1) the sectoral applications and highlight climate relevant information beyond the three user cases, (2) the modeling structure of underlying functions to help users understand the methods and assumptions, i.e. (2.a) the input, (2.b) the mathematical formulae, (3) the definition of abbreviations, acronyms and technical terms. On the presentation side, (3) avoid excessive use of lists, (4) avoid extensive use of links and (5) streamline the case studies.

We appreciate the acknowledgment that "*The toolbox fills a clearly identified gap and could help researchers addressing relevant scientific questions within the scope of EGU*" and the summary of suggestions that we are addressing under each specific comment.

We consider the list of recommendations has been reviewed and answered in previous comments. Below, we provide answers for the specific comments and typing errors:

- **I-39. "*stakeholders*"**: I would appreciate a series of examples for sectoral applications introduced in the beginning of the paper (agriculture, tourism, consumer discretionary stock planning, climate risk for insurer/ infrastructure, energy (wind but also solar/ thermic etc.).

We included a few specific examples:

"There is a strong need and interest for reliable climate forecasts in a wide range of socioeconomic sectors such as energy, agriculture, tourism, health, insurance or logistics to name only a few (White et al., 2017). But the specific information needs for assisting decision-making vary strongly, even within the same sector. For instance, a wind farm owner might be interested in estimating the risk of low cash income due to low winds during a given season and plan a reduction in  production accordingly . This requires local information of near-surface wind speed, combined with the specific performance specifications of the turbines (i.e. relevant wind thresholds vary across wind farms). On the other hand, a grid operator might require country-aggregate information of temperature extremes as a proxy for anticipating electricity demand and ensuring the balance of supply and demand in the electricity grid. Similarly, for the agriculture sector, the required climate information may depend on the specific culture (e.g.: olive, wine, or wheat) and even on the specific crop variety, since each of these crops may have

different phenological evolution, which implies a climate sensitivity to different climate variables and different time periods. This diversity of user needs makes the generation of tailored products costly in time and resources, something that is sometimes known as the last-mile problem of climate services (Celliers et al., 2021)."

Celliers, L., Costa, M., Williams, D., and Rosendo, S.: The 'last mile' for climate data supporting local adaptation, Global Sustainability, 4, E14, doi:10.1017/sus.2021.12, 2021

- **l.41. "*tailored climate information*"**: The transmission channels from climate data to climate relevant information could be slightly more detailed in this section.

Thanks for the comment. We have added the following sentece:

"The generation of tailored climate information can be, for instance, the extraction of global data in a particular region of interest, the correction of the systematic errors that prevent the integration of the climate predictions in impact models or the refinement of the coarse resolution of the climate datasets in order to be representative of the local climate variability."

- **l.50** To address these needs

Thanks for noticing this mismatch. It has been corrected.

- **l.57.** CSTools targets primarily

 Corrected.

- **l.100:** R based

Corrected.

- **l.105-110:** first sentence in the end or footnote (from a detailed description …)

Sentence moved to the end.

- **l.191. "*automatically interpolates all the data onto a common grid*"**: What is the advantage of CST_load (turning ncdf into s2dv_cube), vs. traditional ncdf4 package loading netcdf object directly? In general, key advantages of the package vs others could be better exposed in the paper rather than in the data description vignettes ("Some benefits of using this function are"). In addition, instead of CDO, would it be possible to use internal R functions such as rasterize (package raster)?

We have improved those lines to make the benefits of using CST_Load function much clearer:

"CSTools includes a single but powerful function to retrieve data from netCDF files called CST_Load. This function is a wrapper of the s2dverification Load function which allows to load monthly or daily forecast data together with date-corresponding observations (Manubens et al., 2018). The function allows to easily combine subsets of data stored in multiple files in POSIX file systems or OPeNDAP servers, and is designed to support

custom conventions for distribution of data across files, file naming, and NetCDF structure. Optionally, CSTools can automatically interpolate all the data onto a common grid if necessary, thus greatly removing complexity for the user."

While CST_Load relies on CDO, this reliance is invisible to the user. Other tools could be used by the user to interpolate the data, but the user would have to operate with the data structures returned by CST_Load, i.e. list-objects containing multidimensional arrays with named dimensions for the actual data, and vectors for the longitudes and latitudes. We hope this can already be inferred by a curious reader from the current version of the manuscript, and prefer not to enter into detail.

Note that the aim of the package is to share the methodologies developed by the co-authors, while the function to retrieve data to the R session and the s2dv_cube object are meant for the usability of the package and can be avoided by retrieving the data with other existing packages and using the CSTools functions without prefix CST_.

- **I-196: These sentences: "***Although datasets can be retrieved from OPeNDAP URLs with NetCDF files, in general, the datasets have to be downloaded onto a local repository and formatted to comply with the CST_Load requirements. Observational reference datasets are stored in a folder in separate monthly NetCDF files (other formats are also possible; see https://earth.bsc.es/gitlab/es/s2dverification/-/blob/master/vignettes/data_retrieval.md for more information), while seasonal 200 forecasts are stored by start date in distinct folders (see https://cran.rproject.org/web/packages/CSTools/vignettes/Data_Considerations.html). A python code to download and format the seasonal forecast datasets from the CDS is provided in the repository CDS Seasonal Downloader (https://earth.bsc.es/gitlab/es/cds-seasonal-downloader).*": should be clarified in the paper.

The first part of the text has been removed since function CST_Load is a wrapper of s2dverification package function Load which is described in Manubens et al., 2018. The last sentence in the paragraph has been moved to Section 1 in order to clarify the aim of the package and what is or is not included in the toolbox as early as possible (see the answer to RC#1 comment above 'Data collection, curation and homogenization).

- **I.237. "k-mean": how k is determined? optimal? parametrized?**

The functions CST_WeatherRegimes and CST_EnsClustering include a parameter that should be set by the user indicating the number of clusters or centers. K is not determined by these functions.

- **I.259: Why five methods? can all downscaling methods be used regardless of the climate variable considered? For instance, if a method is developed for surface (10m) wind (e.g. TORRALBA, 2017), can it be applied to humidity, sea-level pressure? If not, the authors could list the best suited input for each method.**

Some of the methods (e.g.: RainFARM) are developed for specific variables and it is specified in the text and also in the new table. Particularly the methods described in Torralba et al. 2017 have been also applied to adjust seasonal forecasts of precipitation and temperature (e.g. Manzanas et al. 2019; Manrique-Suñén et al. 2020). Furthermore, other methods could be used depending on the interest of the user. The methods included in CSTools have been developed or tested under specific conditions although they could be also valid under other assumptions.

- **l.266: Precise applications for each pattern for analog downscaling. What are the main differences, what should we use in which situation? Or is it recommended to use all three and minimize error?**

We have clarified these questions with the following lines:

"Typically, criteria (1) is used to find the analog based on a large-scale variable (e.g . sea level pressure/geopotential in the North Atlantic or sea surface temperature over the tropics). Criteria (2) helps to confirm that both large-scale patterns and the large-scale variable in a local scale (e.g. sea level pressure in the Iberian Peninsula) are consistent. Criteria (3) also measures the similarities between the large-scale variable and a different variable (e.g. surface temperature in the Iberian Peninsula) in the local scale."

- **l.277 and 285: maybe recall the minimal mathematical expression of the effect of orography on downscaling (Terzago et al. 2018)?**

The orographic correction employed in Terzago et al., 2018 introduces the small-scale variability in the downscaled fields deriving it from a reference climatology at fine scale, $c(x,y)$, obtained from long-term time averages of gridded observational precipitation datasets, radar measurements, or from numerical simulations with high-resolution models. In detail, each value of $c(x, y)$ is divided by its local smooth average at the scale of the dataset to be downscaled ($L_o$):

$$w(x,y)=c(x,y) / S[c(x,y)]_{L_o}$$

where $S[ ]_{L_o}$ is a smoothing operator as described in Terzago et al., 2018. The resulting weight field $w(x,y)$ reflects the distribution in space, inside each cell of size $L_o$, of the climatological precipitation in the reference dataset. Notice that this provides a map of weights with both positive and negative values and that, on average, precipitation at scale $L_o$ is conserved using this approach. The weights are then applied to the fine-scale field produced by the RainFARM procedure generating a new field in which precipitation is reduced or intensified according to the weights obtained from the long-term climatology. As a last step, the final amplitude is adjusted to conserve average precipitation at scale $L_o$.

This procedure is quite long to be explained in the text, so we prefer not to report this level of detail in the manuscript. In the manuscript, we added a sentence to better explain the type of orographic correction applied.

Previous version (l.277): "and recently improved for regions with complex orography (Terzago et al., 2018)."

New version: "and recently improved for regions with complex orography, for which the fine-scale fields produced by RainFARM are corrected using weights derived from a fine scale precipitation climatology (Terzago et al., 2018)."

- **l.291."C*ST_AnalogsPredictors function downscales precipitation or maximum/minimum temperature low resolution forecast output data, in a domain centred over Iberian Peninsula*". The function "Analogs Predictors" works in Spain only?**

The function was initially developed for downscaling global model outputs over the Iberian Peninsula. Predictors, metrics for selecting analogs, observational data for calibration and other options were tuned and tested for this region (Amblar et al., 2020; Hernanz et al., 2021a; Hernanz et al., 2021b). The function could be used in other regions, but bearing in mind that the same collection of predictors will be used and that format of observational data is respected. There are new future plans to upgrade this function allowing more flexibility in data formats and selection of options. The previous mention of 'in a domain centered over the Iberian Peninsula' has been removed.

Amblar-Francés, M. P., Ramos-Calzado, P., Sanchis-Lladó, J., Hernanz-Lázaro, A., Peral-García, M. C., Navascués, B., Dominguez-Alonso, M., Pastor-Saavedra, M. A., and Rodríguez-Camino, E.: High resolution climate change projections for the Pyrenees region, Adv. Sci. Res., 17, 191–208, https://doi.org/10.5194/asr-17-191-2020, 2020.

Hernanz, A., García-Valero, J. A., Domínguez, M., Ramos-Calzado, P., Pastor-Saavedra, M. A. and Rodríguez-Camino, E. (2021a). Evaluation of statistical downscaling methods for climate change projections over Spain: present conditions with perfect predictors. *International Journal of Climatology*, 42( 2), 762– 776. https://doi.org/10.1002/joc.7271

Hernanz, A., García-Valero, J. A., Domínguez, M., & Rodríguez-Camino, E. (2021b). Evaluation of statistical downscaling methods for climate change projections over Spain: Future conditions with pseudo reality (transferability experiment). *International Journal of Climatology*, 1– 14. https://doi.org/10.1002/joc.7464

- **l.292: in a domain centered over Iberian Peninsula**

Removed.

- **l. 309: better explain (1) calibration methods (evmos, mse_min, crps_min, rpc-based), (2) on what variables / conditions should the choice of the method be based?**

The choice of the method depends on the metric that the user wants to improve.

For example, if a user wants to improve the quality of a deterministic product, "mse_min" could be a good option (as it tries to reduce the error). On the other hand, if the user

wants to improve the quality of a probabilistic product, "crps_min" may be a better option (as it tries to reduce the CRPS).

Also, although the forecast quality is improved after calibration when measured with one metric, it may have been worsened when measured with another metric. Therefore, the choice depends on the user's needs.

We hope that the new table 1 helps in regards to understanding each calibration method. Furthermore, we have added a sentence to help understand the general idea:

"CST_Calibration performs the correction on the forecast systems' simulations using five different member-by-member methodologies, where each methodology can adjust one or more statistical properties of the predictions. The selection of the most appropriate method will thus depend on the user's needs."

- **Visalisation: Maybe insert some "visualization" (i.e. output of the functions described for each of them so we know what it does, even if it's in the next section)?**

New figures have been included.

- **l.399: "Oops, ha ocurrido un error 404 La página a la que intentas acceder al parecer no existe o ha sido eliminada de nuestro sitio web"**

Removed.

- **l.445. Code box. please add a link where to find the data file referred to in the link to improve reproducibility**

Most of the data was downloaded from the CDS using the code described in Section 2.2.1 and we have added the link to the CHIRPS datasets to the text that can be downloaded from https://data.chc.ucsb.edu/products/CHIRPS-2.0/ while data information is provided in https://www.chc.ucsb.edu/data. We don't include the original data along with the manuscript since they are already freely available. Instead we provide a new appendix B with details on data collection, curation, homogenization, and requirements for CST_Load. We have revised the manuscript to make clear that the aim of CSTools is to bring methods to post-process climate forecasts and a framework to develop climate services analysis but not to provide data.

- **Figure 3: isn't the density shape giving a somewhat misguiding idea of the underlying distribution ("smoother" than it is)? Apart from that this figure is very nice.**

The underlying distribution is unknown and its density function is approximated by dressing a limited number of ensemble members. The "ensemble dressing" is performed here by using the Kernel Density Estimate technique with a gaussian kernel (Bröcker and Smith 2008). Silverman's rule of thumb is used to select the spread of the kernel,

which controls the degree of smoothing. This information has been included in the description of PlotForecastPDF in section 2.2.6.

Bröcker, J., & Smith, L. A. (2008). From ensemble forecasts to predictive distribution functions. In Tellus A (Vol. 60, Issue 4, pp. 663–678). Informa UK Limited. https://doi.org/10.1111/j.1600-0870.2008.00333.x

- **SNOWPACK inputs: first introduced line 550, while the inputs of the models are introduced line 714 (consider restructuring)**

The text has been restructured.

- **l.652-663: very important issue in climate data manipulation: the size of the data. I think a full subsection could be dedicated to this topic in the section 2, and then simply referred to in the case study section where we want to focus on the application side (and not the technical issue).**

This topic is indeed important but it is too complex to be included in the manuscript. We consider that use case 2 is a good opportunity to show the impact of downscaling on the data size as well as the impact of the parameter *nf* in the RainFARM method. In section 2.2.1, we proposed to use the package startR when the RAM memory is exceeded by the size of the dataset.

**Appendix B: Details on data collection, curation, homogenization, and requirements for CST_Load**

In order to use CST_Load, the storage needs to be homogenized. CST_Load accepts several parameters to configure the loading and interpolation of data. The CST_Load documentation in the reference manual is linked to the s2dverification (Manubens et al., 2018) reference manual where the description of all parameters is detailed.

Basically, CST_Load requires path patterns pointing to the NetCDF files or OPeNDAP URLs requested via the other parameters. A variable with a matching name must be present in the files. The path patterns, one for each experimental/observational dataset to be loaded, express the set of files comprising each dataset. Therefore, a path pattern is a string containing some specific wildcards that are recognised and replaced by the corresponding values by CST_Load. The most commonly used wildcards in a path pattern specification are "$START_DATE$", "$STORE_FREQ$", and "$VAR_NAME$". For example, given a dataset that consists of the following files:

- /data/datasetA/monthly/tas_20180101.nc
- /data/datasetA/monthly/tas_20180201.nc
- /data/datasetA/monthly/tas_20180301.nc

The path pattern to express the set of files would be as follows:

"/data/datasetA/$STORE_FREQ$/$VAR_NAME$_$START_DATE$.nc"

The use case 1, "Assessing the odds of an extreme event", directly loads wind speed (in surface for the case of SEAS5 and at 100m for the case of ERA5). However, this variable is not directly available in the Copernicus CDS while u and v wind components are in 6 hourly and monthly frequencies. In order to get the monthly wind speed, the 6 hourly frequency components are used to calculate the 6 hourly wind speed and, then, calculate the monthly average of the wind speed using CDO (Schulzweida, 2019). Notice that averaging the monthly wind components may lead to a different result. To automatise this calculation on all the files in a folder, the following bash code could be adapted:

```
path_output="./data/monthly/sfcWind_f6h/"

path_component="./data/6hourly/"
for year in {1993..2018}
do
for month in 01 02 12
do
output_file=${path_output}"sfcWind_${year}${month}01.nc"
uas_file=${path_component}"uas/uas_${year}${month}01.nc"
vas_file=${path_component}"vas/vas_${year}${month}01.nc"
cdo -L -b F64 -f nc -setunit,'m/s' \
  -setname,"sfcWind" \
  -sqrt \
   -add \
     -mul -selname,uas $uas_file -selname,uas $uas_file \
     -mul -selname,vas $vas_file -selname,vas $vas_file \
  ${output_file}_tmp
```

```
cdo monmean ${output_file}_tmp $output_file

done
done
```

An equivalent script, using CDO dailymean operator, can be used to convert the data downloaded into daily mean values for use case 2 and 3.

---

## Author Comment (AC2)

**Answer to RC2: 'Comment on gmd-2021-368', Anonymous Referee #2, 10 Jan 2022**

Dear Referee,

Thanks for your comments on our manuscript. We really appreciate them and also that you took the time to run the examples available on CRAN. We are glad to read that you consider it a valuable contribution.

As deserved, below we provide a detailed answer to each of your comments.

Kind regards,

Núria Pérez-Zanón
On behalf of all manuscript authors

> In summary, I believe this tool, even though not provide any new modelling option (but this was not the aim of the authors), is a valuable contribution and has good potential to impact several sectoral applications. Nevertheless, I believe some more details must be provided to make it really accessible to the wider public (i.e., even stakeholders not particularly expert in forecasting issues) and, in some cases, even experts. In particular, I refer to the data retrieval and formatting section, which could be very "labour-intensive" as the same authors state. All the examples provided use either link to static paths (in the paper) or already pre-processed input data (vignettes). I suggest the authors go more into detail on that and provide at least one example starting from raw data.

We added a new Appendix detailing the process followed to download and homogenize the storage for the first use case based on the code provided to download and locally store the seasonal forecast data from Copernicus CDS.

However, while the Copernicus Climate Data Store is currently one of the main sources of climate datasets, there exist other data repositories (e.g.: National Center for Environmental Information NCEI). The different data repositories can deliver the datasets in different formats for both file and structure, making it challenging to create a single software/function that considers all requirements.

In section 1.2, we have added a new first step to the list of steps to explain the climate forecast post-processing chain:

• Data collection, curation and homogenization: This includes collection of data from heterogeneous remote data sources, storage and indexing into local or

organisation-accessible file systems or servers, and homogenization for all data files to comply with common internal conventions. The complexity of this step can be high, particularly if the data sources do not follow community standards. This step is out of the scope of this manuscript and the CSTools toolbox, and the use of other tools such as the cds-data-downloader (https://earth.bsc.es/gitlab/es/cds-seasonal-downloader) is suggested for this purpose.

In the same section, we also clarify that the main purpose of the package is not data retrieval:

The primary aim of CSTools is, therefore, to share post-processing methods (i.e. correction methods for forecast calibration, classification methods for multi-model forecast combination or scenario selection, downscaling methods, and visualization tools) that aren't currently available in other software packages or, whether the method exists in separate software, their inclusion facilitates the comparison of the results. Because additional steps are required (i.e. data retrieval from remote servers, storage and, indexing into local or organization-accessible file systems or servers, curation and formatting, and finally loading from the file systems or servers onto RAM memory of the processing machines), we provide extra functions and scripts in order to facilitate the use of the toolbox.

Still concerning input data, another common feature of the examples offered is that they seem to rely only on global/large scale gridded datasets. In my experience, I've learned that such datasets often don't fit adequately ground observations for specific regions. If the monitoring network (e.g. rain gauges) is dense enough, it can be used in turn to prepare one's own high-resolution (let's say) dataset. It's not clear to me if/how such datasets can be included, for example for correction or validation purposes.

Since the climate forecasts are global gridded datasets, most of the applications are built on references that are also gridded datasets, such as reanalyses. For correction or validation purposes, CSTools methods could be used to post-process a climate forecast with is-situ observations. The important step would be to create two data arrays, one for the climate forecast and another for the in-situ observations, that match the temporal and spatial dimensions: by selecting the closest grid point, by averaging observations within one gridpoint, or even, by regridding the climate forecast to better select the corresponding gridpoint. Other considerations, like the spatial representativeness of the in-situ observations, should be taken into account.

Another comment concerns the structure of the three use cases provided. I suggest describing them more homogeneously and streamlining them. The third use case is a bit sacrificed, in my opinion.

We have homogenized the text of the three use cases by following the suggested scheme of Referee #1:   (i) application (why?), (ii) data required/ input and at what

resolution / frequency, (iii) process required from source to model, (iv) code guide, (v) output and final data visualization and (vi) interpretation.

> Finally, I suggest organizing better (in a more straightforward way) the connection between functions developed and corresponding literature references, to support the user in going into details with the theoretical aspects behind them. Maybe, some synoptic tables (even as an appendix), in addition to existing text, could help.

Following this comment and others from Referee #1, we have updated table 1 to include a description and references for each function.

[revised manuscript text omitted]

Below I provide some specific comments (and highlight some typos). I recommend careful re-reading of the manuscript. I hope my review helps improve the overall quality of the manuscript and makes more accessible the interesting toolbox developed.

We acknowledge this detailed review and we answer each comment below.

L 66: as illustrated in Fig. 1

Corrected.

L100: R-based

Corrected.

L104: please check this sentence

The sentence "CSTools could nonetheless be useful to research scientists, as it is made compatible some of the aforementioned R packages." has been corrected as "CSTools could nonetheless be useful to research scientists, as it has been designed to be compatible with some of the aforementioned R packages."

L130: maybe "each function"?

Corrected, thanks.

L191: to automatically interpolate

Corrected.

L193: lotlan_data for temperature? Please check

The name of the data object could be changed to something similar to "lonlat_temp" which would be more appropriate to reflect the fact that it is a temperature data sample and more coherent with the lonlat_prec data sample name. To fix this problem, we have opened an issue in the gitlab repository (https://earth.bsc.es/gitlab/external/cstools/-/issues/84) and included this change in the package.

L197: downloaded into (or simply "in")

This line has been removed.

L244: "The amount of categories can be changed and are taken as…" please check this sentence. To which subject is the verb "are" referred? To the categories?

The sentence has been checked.

Previous version: "The amount of categories can be changed and are taken as the climatological quantiles (e.g. terciles), extracted from the observational data."

New version: "The user can set up the total number of categories that will be used to define the observed climatological quantiles."

L291: not clear: is this function available only for the Iberian Peninsula? Will it be available for other areas in the future?

The mention of the Iberian Peninsula has been removed since it was originally developed for the Iberian Peninsula and generalized in CSTools to accept inputs for any region in which high-resolution observational datasets are available.

L301: not clear: here, too, is this function available only for NAO?

We have re-written these lines as follows:

Sánchez-García et al. (2019) used the North Atlantic Oscillation (NAO) to improve the skill of the seasonal precipitation forecast over the Iberian Peninsula. Given that this methodology could be explored to improve the skill of different climate variables that are led by other climate indices, the method has been generalized and named Best Estimate Index (BEI).

L375: A comparison … IS also possible

We thank the reviewer for finding out this error. Corrected.

L386: three example case studies

Corrected.

L399: the link does not work. However, I would prefer some more technical link than that to a newspaper

The link has been removed.

L401: I guess IP stands for Iberian Peninsula. But his term is used only some words before, so please check the sentence and rephrase

We have fixed it.

Previous version: Very high wind speeds were later recorded over large part of the Iberian Peninsula due to 4 cyclones going across the IP (AEMET, 2018).

New version: Very high wind speeds were later recorded over large parts of the Iberian Peninsula due to the passing of four cyclones (AEMET, 2018).

L453: by?

Removed.

L503: "only one member": it's better to tell how many members make up the ensemble

We completely agree with this comment and have added the clarification "out of 25" to the text.

L509: please explain what "ensemble dressing" means.

We have described the ensemble dressing procedure in section 2.2.6, and referenced that section in this part of the manuscript.

L545: I would write "agriculture and industry, while meltwater shortage …"

The suggestion has been included in the text.

L597: "the result is" (better) or "the results are"

While reviewing the text, this line has been re-shaped to: "The monthly spectral slopes obtained". Thanks for noticing.

Figure 6a: I guess this map shows one of the 25 possible precipitation fields for 11 December 1993 given by the SEAS5 ensemble

Yes, you are right. We have corrected it as follows: "Figure 6: One out of the 25 ensemble members of the original ensemble".

L719 (and elsewhere): please check throughout the text if there are shifts using tenses (from the present to the past and vice versa)

We appreciate this comment and we have revised the text while re-structuring the use cases.

LL719-720: not clear if these operations were made through CSTools (please refer also to main comments)

At the time of writing this manuscript, these operations were done using other software. However, the functions CST_RFTemp could be used to post-processing the temperature dataset and CST_Load (which allows regridding with CDO) could be used to do the bilinear interpolation of the rest of the variables.

L723: "the SNOWPACK model is run for each of the 21 seasonal forecasts over the hindcast period 1996-2016". Only here the objective of the use case is clearly stated. I suggest declaring it at the beginning of the section.

We have re-written the use case and added the following sentence at the beginning to explain the objective:

"The post-processing of seasonal precipitation forecast in the Alps to be used as input for the SNOWPACK model is shown in this use case, as well as, the result of the SNOWPACK model snow depth."

L741: again, for what period? State clearly the objectives of the exercise at the beginning of the section.

Given the rewritten of the use cases following the previous reviewer's suggestion, we consider the information is now clear.

L794: at the end of this section, I realize that the fact that the SCHEME hydrological model is used is not so relevant, after all. The case study could be generalized to any (semi-distributed or even distributed) hydrological model requiring precipitation and temperature forecasts.

Indeed. The use case is post-processing temperatures and precipitation seasonal forecasts. We have re-written the use case to follow the structure suggested by reviewer #1 but also to clearly show the target users of this use case.

L804: "(see e.g. Fig. 4)" I would remove this test in brackets.

Removed.

L813: also, agricultural issues are involved (drought, irrigation needs, water resources management, etc.)

The sentence has been modified to include this comment as:

"Similarly, these use cases are relevant for risk management of high wind speed, coastal and flooding events, as well as, agricultural issues implied by droughts, irrigation needs or water resources management."

L815: what about the other features? I think this sentence underestimates other aspects of the tool. Please explain in more detail.

We really appreciate this comment. We have improved the sentence to highlight that apart from the methods, other aspects of the software are valuable.

Please note: Appendixes A and B are not referred to in the main text. They should be and contextualized.

We appreciate the careful review. Appendices are now cited in the text.

**Appendix B: Details on data collection, curation, homogenization, and requirements for CST_Load**

In order to use CST_Load, the storage needs to be homogenized. CST_Load accepts several parameters to configure the loading and interpolation of data. The CST_Load documentation in the reference manual is linked to the s2dverification (Manubens et al., 2018) reference manual where the description of all parameters is detailed.

Basically, CST_Load requires path patterns pointing to the NetCDF files or OPeNDAP URLs requested via the other parameters. A variable with a matching name must be present in the files. The path patterns, one for each experimental/observational dataset to be loaded, express the set of files comprising each dataset. Therefore, a path pattern is a string containing some specific wildcards that are recognised and replaced by the corresponding values by CST_Load. The most commonly used wildcards in a path pattern specification are "$START_DATE$", "$STORE_FREQ$", and "$VAR_NAME$". For example, given a dataset that consists of the following files:

- /data/datasetA/monthly/tas_20180101.nc
- /data/datasetA/monthly/tas_20180201.nc
- /data/datasetA/monthly/tas_20180301.nc

The path pattern to express the set of files would be as follows:

"/data/datasetA/$STORE_FREQ$/$VAR_NAME$_$START_DATE$.nc"

The use case 1, "Assessing the odds of an extreme event", directly loads wind speed (in surface for the case of SEAS5 and at 100m for the case of ERA5). However, this variable is not directly available in the Copernicus CDS while u and v wind components are in 6 hourly and monthly frequencies. In order to get the monthly wind speed, the 6 hourly frequency components are used to calculate the 6 hourly wind speed and, then, calculate the monthly average of the wind speed using CDO (Schulzweida, 2019). Notice that averaging the monthly wind components may lead to a different result. To automatise this calculation on all the files in a folder, the following bash code could be adapted:

```
path_output="./data/monthly/sfcWind_f6h/"

path_component="./data/6hourly/"
for year in {1993..2018}
do
for month in 01 02 12
do
output_file=${path_output}"sfcWind_${year}${month}01.nc"
uas_file=${path_component}"uas/uas_${year}${month}01.nc"
vas_file=${path_component}"vas/vas_${year}${month}01.nc"
cdo -L -b F64 -f nc -setunit,'m/s' \
  -setname,"sfcWind" \
  -sqrt \
    -add \
      -mul -selname,uas $uas_file -selname,uas $uas_file \
      -mul -selname,vas $vas_file -selname,vas $vas_file \
  ${output_file}_tmp
```

```
cdo monmean ${output_file}_tmp $output_file

done
done
```

An equivalent script, using CDO dailymean operator, can be used to convert the data downloaded into daily mean values for use case 2 and 3.

---

## Author Response (AR2)

**Answer to Report #1 (Submitted on 30 Apr 2022) by Anonymous referee #2**

**For final publication, the manuscript should be accepted as is**

> **Suggestions for revision or reasons for rejection (will be published if the paper is accepted for final publication)**
> I am satisfied with the responses to my comments. I just wonder if the considerable dimensions of the new Table 1 do not lead the authors to consider inserting it in the Appendix. I would do so, leaving some more synthetic information in Table 1.

Thanks for the comment. Yes, we considered inserting it as an appendix. However, since Table 1 seemed relevant for both referees, we had originally decided to keep it as part of the main manuscript. Given this comment and a comment from Referee #3 on the readability of the manuscript, we have followed this suggestion and moved the current version to the appendix and synthesized Table 1 as follows:

Table 1. Summary of the functions and methods by category included in CSTools. Prefix "CST_" refers to functions working on a specific object class called "s2dv_cube". Asterisk indicates functions that are used in vignettes (see Appendix B for a detailed table).

| | |
|---|---|
| **Retrieval and transformation** | CST_Load*, CST_Anomaly*, CST_SaveExp, CST_MergeDims, CST_SplitDims, as.s2dv_cube, s2dv_cube |
| **Classification** | CST_MultiEOF, CST_WeatherRegimes*, CST_RegimesAssign*, CST_CategoricalEnsCombination, CST_EnsClustering* |
| **Downscaling** | CST_Analogs*, CST_RainFarm*, CST_RFTemp, CST_AdamontAnalog, CST_AnalogsPredictors |
| **Correction** | CST_BEI_Weighting*, CST_BiasCorrection, CST_Calibration, CST_QuantileMapping, CST_DynBiasCorrection |
| **Assessment** | CST_MultiMetric*, CST_MultivarRMSE* |
| **Visualization** | PlotCombinedMap*, PlotForecastPDF*, PlotMostLikelyQuantileMap*, PlotPDFsOLE, PlotTriangles4Categories* |

**Answer to Report #2 (Submitted on 12 May 2022) by Referee #3, Matteo De Felice**

**accepted subject to minor revisions**

> **Suggestions for revision or reasons for rejection (will be published if the paper is accepted for final publication)**
> This paper describes an impressive software package in terms of features. Given the complexity and the number of features provided by the package, the paper results a bit long and hard to follow. In general, I would suggest the authors to shorten the paper possibly:
> - Reducing section 2.1, especially the first half
> - Keeping only one use case and moving the other two to the supplementary

1. Reducing Section 2.1
We suspect the reviewer means Section 2.2.1 and not 2.1, since Section 2.1 is only 3 paragraphs long and describes the data structure of the package (which is essential in our opinion).

On the other hand, based on this comment and a similar comment from the other referee, we have reduced Table 1 and moved the current version to an appendix. Furthermore, the first paragraph in Section 2.2.1 has been removed.

2. Moving the use cases
We would like to keep all three use cases in the main document if possible. CSTools already has a GitLab project and a CRAN repository where a reference manual documents all available functions. Rather than documenting the toolbox, the main purpose of this manuscript is to showcase the potential usability of CSTools in a climate service context. Therefore, we consider that keeping just one use case in the main manuscript would lead readers to underestimate its flexibility.

> Here, some additional comments:
> - The authors mention EUROSIP in the first paragraph but it has been taken over by the C3S Multi-model system (see https://www.ecmwf.int/en/about/media-centre/news/2019/c3s-multi-model-seasonal-forecasting-system-takes-over-eurosip). I would suggest mention this one.

Changed.

> - After the list of the categories at the beginning of Section 1.2, I would specify which ones are covered by CSTools

All the items of the list are covered by CSTools, except for the first one. The subsequent paragraph after the list of categories (beginning of Section 1.2) has been modified to:

"The primary aim of CSTools is therefore to make post-processing methods (i.e. **correction methods for forecast calibration, classification methods for multi-model forecast**

**combination or scenario selection, downscaling methods, and visualization tools**)
available in one coherent framework in order to facilitate analysis or the post-processing of
data such that might be required by an impact model.  Because additional steps are
required, CSTools also includes functions for **data retrieval and formatting as well as skill
assessment** in order to facilitate the use of the toolbox.*"*
* * *
- The authors write "CSTools, on the other hand, targets scientists interested in providing a
climate product to some final users" but this sentence is unclear, what are exactly the
differences between CSTools and other tools that make it more suitable for "final users"?
* * *
The underlined sentence was added: "The main purpose of these different packages is the
facilitation of research. CSTools, on the other hand, targets scientists interested in providing
a climate product to some final users. This is done by allowing the creation of complete
post-processing chains, from data retrieval to the creation of high-quality datasets to feed
impact models or tailored forecast visualization products."
* * *
- The authors state that CSTools is compatible with other packages, this point should be
better explained.
* * *
CSTools functions operate on data array objects. The array class is the same class used by
other packages such as s2dverification, SpecsVerification, easyVerification and startR.
Furthermore, CSTools is also compatible with CSIndicators as the latter accepts s2dv_cube
objects as inputs directly. If a package operates on a different class of objects (e.g.
data.frame), some transformations will be required, thus breaking the compatibility. The
following sentence has been adjusted:

"The CSTools development guidelines have been designed to maximise compatibility with
other libraries such as s2dverification, s2dv, SpecsVerification, easyVerification and startR,
all of them designed to operate fundamentally with the same array class. Furthermore,
CSTools is also compatible with CSIndicators (Pérez-Zanón et al., 2021) as the latter
accepts "s2dv_cube" objects as inputs."
* * *
- In Line 192 some 'common guidelines' are mentioned: does it mean that they published
specifications/requirements to follow to implement - for example - another input data format
or another post-processing method?
* * *
The common guidelines, which are in the supplementary material, is a document that
explains the most relevant aspects to consider when adding a new function to CSTools:
avoid duplication of methods already included in other software tools without justification,
data formats and examples. We have added a reference to the supplementary material in
the main text (2nd paragraph of Section 2).
* * *
- What is exactly the repository of CSTools? The repository at
https://earth.bsc.es/gitlab/external/cstools says that is the repository of the MEDSCOPE
project.

Thanks for pointing this outdated file in the repository. Yes, this is the CSTools repository. The README file has been updated to clarify this point and to include the reference to this manuscript, the link to the CRAN repository and some other relevant information.

> - I would add a paragraph in section 2.1 giving some more details on the possibility to do lazy and distribute calculations using startR, that is a very important topic for climate scientists

We feel adding explanations on how to distribute calculations using startR would be lengthy and detract from the main point of that section. However, we agree that distribute computing is a very important issue, so instead we have added the following sentence pointing to the startR gitlab repository where one can find an example of a CSTools calibration method integrated in a startR workflow:

"An example on how to use a CSTools function in a startR workflow can be found in its GitLab repository (https://earth.bsc.es/gitlab/es/startR)."

> - The fourth column in the Table 1 is a bit confusing, what is its meaning? For 's2dverification' I assume that its functions are directly called into CSTools code, but what about the rows with empty values ('-')? What's the difference between '-' and 'adaptation to CSTools'?

To improve the readability of the main manuscript, table 1 has been moved to Appendix B and substituted in the main text by a synthetic table:

Table 1. Summary of the functions and methods by category included in CSTools. Prefix "CST_" refers to functions working on a specific object class called "s2dv_cube". Asterisk indicates functions that are used in vignettes (see Appendix B for a detailed table).

| | |
|---|---|
| **Retrieval and transformation** | CST_Load*, CST_Anomaly*, CST_SaveExp, CST_MergeDims, CST_SplitDims, as.s2dv_cube, s2dv_cube |
| **Classification** | CST_MultiEOF, CST_WeatherRegimes*, CST_RegimesAssign*, CST_CategoricalEnsCombination, CST_EnsClustering* |
| **Downscaling** | CST_Analogs*, CST_RainFarm*, CST_RFTemp, CST_AdamontAnalog, CST_AnalogsPredictors |
| **Correction** | CST_BEI_Weighting*, CST_BiasCorrection, CST_Calibration, CST_QuantileMapping, CST_DynBiasCorrection |
| **Assessment** | CST_MultiMetric*, CST_MultivarRMSE* |

| Visualization | PlotCombinedMap*, PlotForecastPDF*, PlotMostLikelyQuantileMap*, PlotPDFsOLE, PlotTriangles4Categories* |
|---|---|

The fourth column of the table (now in Appendix B), aims to distinguish methods that are novel software developments included in CSTools from functions that are wrappers from other packages and existing software codes that have been adapted or re-coded to be included in CSTools. The original title of this column, 'Original code version', has been changed to 'Original development?'. The following possible expressions are used to fill the column:

- Yes: meaning that the function is a novel development
- Adaptation to CSTools: a version of the method is already available as a software function or software code but has been included in CSTools following the development guidelines. This is the case, for example, of the RainFARM downscaling method that exists as standalone Julia and R packages but has been adapted here to be part of CSTools.

The table has been reviewed and all the cells are now filled.

> - Are the specifications of s2dv_cube described? Would it be possible for someone to create a function generating s2dv_cube objects that can be used straightly into CSTools?

Yes, the detailed description of the s2dv_cube is included in the supplementary material.

There is already a function generating s2dv_cube objects called 's2dv_cube' (see 2nd paragraph of Section 2.2.1).

> - I would suggest removing the 'single but powerful' at the beginning of Line 280, I understand the enthusiasm but it seems a bit an exaggeration.

The expression has been removed.

> - Does the function CST_MultiEOFs deal with ensembles (i.e. working directly with the members without using the ensemble mean)? How?

The CST_MultiEOF function works on multiple variables (e.g. geopotential at different levels, geopotential and temperature etc.). Typically it is applied separately to timeseries of individual ensemble members and not to the ensemble mean (applying it to the ensemble mean would also suppress a large fraction of the variability). In the present implementation it is applied to a series of fields in time (considering together both the forecast time and the starting dates), producing separate sets of EOFs and PCs for each other ensemble dimension. So if an ensemble of experiments with different starting dates and forecast times is provided for analysis, the result will be EOFs and PCs computed separately for each ensemble member.

The description in the manuscript has been adjusted as follows:

"The CST_MultiEOF function allows conducting Empirical Orthogonal Functions (EOF) analysis simultaneously over multiple variables for either each ensemble member or all the ensemble members concatenated altogether (i.e., it can be applied to each one of the ensemble members separately or to the whole ensemble)."

In general, I think the paper is relevant and well-done but, however, before publication I think it should be simplified to improve the readability, especially for non-climate scientists.

Thanks for your feedback. We really appreciate the time you dedicated to help us improve the readability.

---

## Author Response (AR3)

Dear editor Dr. Jinkyu Hong,

Thanks for noticing this undesired format. We have removed the subtitles 'Code Step X' and included them as a commented line in the code boxes as '# Step X' as they are mentioned in the text and in figure 10.

We look forward to your answer.

Kind regards,

Núria Pérez-Zanón
On behalf of all manuscript authors